# Targeted sampling of natural product space to identify bioactive natural product-like polyketide macrolides

Darryl M. Wilson[1], Daniel J. Driedger[1], Dennis Y. Liu[1], Sandra Keerthisinghe[2], Adrian Hermann [3], Christoph Bieniossek[3], Roger G. Linington [1,2] ✉ & Robert A. Britton [1] ✉

Polyketide or polyketide-like macrolides (pMLs) continue to serve as a source of inspiration for drug discovery. However, their inherent structural and stereochemical complexity challenges efforts to explore related regions of chemical space more broadly. Here, we report a strategy termed the Targeted Sampling of Natural Product space (TSNaP) that is designed to identify and assess regions of chemical space bounded by this important class of molecules. Using TSNaP, a family of tetrahydrofuran-containing pMLs are computationally assembled from pML inspired building blocks to provide a large collection of natural product-like virtual pMLs. By scoring functional group and volumetric overlap against their natural counterparts, a collection of compounds are prioritized for targeted synthesis. Using a modular and stereoselective synthetic approach, a library of polyketide-like macrolides are prepared to sample these unpopulated regions of pML chemical space. Validation of this TSNaP approach by screening this library against a panel of whole-cell biological assays, reveals hit rates exceeding those typically encountered in small molecule libraries. This study suggests that the TSNaP approach may be more broadly useful for the design of improved chemical libraries for drug discovery.

Among the families of natural products that continue to inspire modern drug discovery, macrocycles are preeminent[1]. Macrocycles occupy unique regions of chemical space that enable them to target proteins that are conventionally inaccessible with small molecule drugs, including low druggability targets such as protein-protein interactions[2]. As a testament to this, approved macrocyclic drugs see broad applications as immunomodulators, antibiotics, and anthelmintics[3]. Yet, biologically active macrocycles often show poor adherence to conventional rules (e.g. rule of five, polar surface area) governing drug-likeness. To address this issue Villar et. al. performed detailed analyses of protein-macrocycle binding and established a set of design criteria for creating pharmaceutically relevant macrocycles[4]. Critically, this seminal study established that biologically active macrocycles share three key conserved features: embedded degrees of unsaturation (e.g. heterocycles, double bonds) within the macrocycle, single heavy atom peripheral groups (e.g. methyl, hydroxyl, chloro) that decorate the macrocycle, and multi-atom side chains (e.g. heterocycles, esters) that are attached to the macrocycle[4]. Together these features offer a three-pronged approach to protein binding, where (i) degrees of unsaturation modulate the entropic penalties associated with binding; (ii) peripheral groups bind complimentary functionalities on a protein surface; and (iii) side chains serve to bind surfaces of

[1]Department of Chemistry, Simon Fraser University, Burnaby, BC V5A 1S6, Canada. [2]Center for High-Throughput Chemical Biology, Simon Fraser University, Burnaby, BC V5A 1S6, Canada. [3]Roche Pharma Research and Early Development, Roche Innovation Center Basel, F. Hoffmann-La Roche Ltd, Grenzacherstrasse 124, 4070 Basel, Switzerland. ✉e-mail: rliningt@sfu.ca; rbritton@sfu.ca

proteins or adjacent clefts and pockets (Fig. 1A)[4]. Additional conformational studies have shown that macrocycles are also capable of burying polar functional groups, exhibiting "chameleonic" behavior which serves to modulate their cellular permeability and physicochemical properties[5–7]. Macrocycles along with many other natural products may also benefit from active transport mechanisms[8]. Yudin and co-workers have reviewed the impact of structure on macrocycle conformation[9], and recently disclosed several remarkable examples where metastable conformations of macrocyclic peptides were locked out by utilizing a dominant rotor approach[10]. Given the various binding

and conformational intricacies associated with macrocycles, it is unsurprising that they have been referred to as the "smallest examples of biomolecules", with functional sub-domains that modulate potency, physicochemical, and pharmacokinetic properties[3].

Macrocycles encompass many dissimilar chemotypes related only by the presence of a large ring, yet most approved macrocyclic drugs trace their origins to natural products[3]. Thus, it is helpful to view macrocycles as being comprised of distinct structural subclasses with differing biosynthetic origins or dominant functionalities within the macrocycle (e.g. polyketide or polyketide-like macrolides (pMLs),

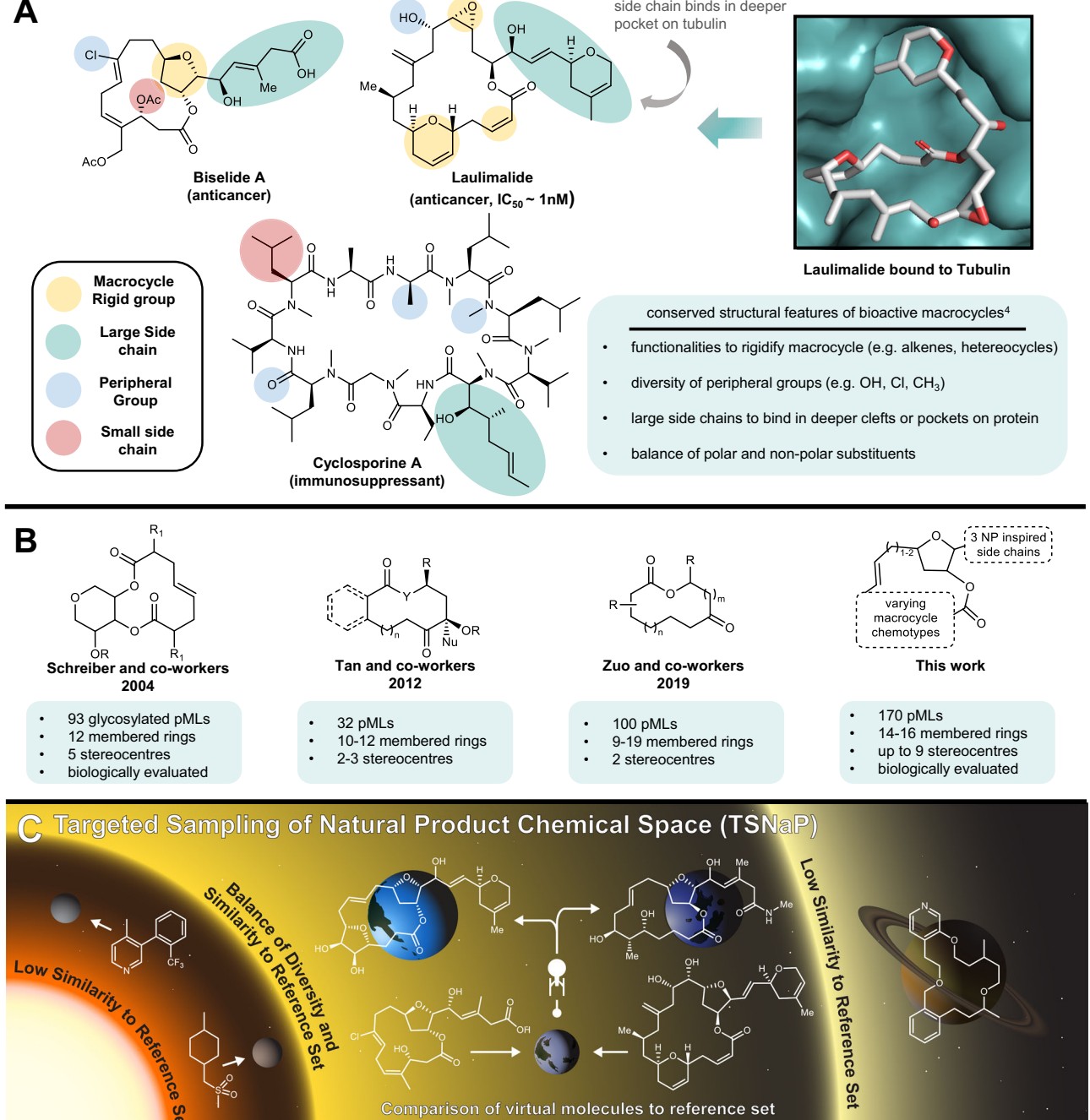

**Fig. 1 | Inspiration and overview of the TSNaP approach. A** Several bioactive macrocycles with select conserved structural features highlighted including rigidifying groups (yellow), large side chains (teal), small side chains (red), and peripheral groups (dark blue). The way these structural features impact protein binding is exemplified in an X-ray crystal structure of the polyketide macrocyclic lactone laulimalide bound to tubulin. **B** Precedent for the construction of de novo pML libraries and this work. **C** The TSNaP approach involves comparison of a large virtual collection of molecules to a reference set of compounds enabling navigation of natural product (NP) relevant space.

macrocyclic peptides, non-ribosomal peptide synthetase (NRPS)-derived macrolactams, or hybrid polyketide-NRPS derived macrocycles). Advances in technologies such as DNA templated synthesis[11], phage display[12], and mRNA display[13] have enabled the synthesis of vast libraries of macrocycles. However, these methods focus on the synthesis of highly nitrogen-enriched macrocyclic peptides and macrolactams. Yet, pMLs represent 6% of all microbial secondary metabolites and are disproportionately represented among approved macrocyclic drugs[3,14,15]. The scarcity of research on non-peptidic macrocycles has not gone unnoticed by researchers[16,17], with some pointing out that the current landscape results from ready availability of amino acids as peptide precursors and specialized technologies suited to the synthesis of cyclic peptides (e.g. solid phase peptide synthesis)[16].

Unlike macrocyclic peptides, many of the elaborate precursors necessary for the construction of pMLs are not commercially available and require lengthy custom syntheses. Moreover, the highly oxygenated scaffolds of pMLs necessitate extensive redox and protecting group manipulations, which limits the technologies suited to their synthesis. Owing to these limitations, pML libraries tend to be constructed using diversity-oriented-synthesis strategies and are comparatively limited in size and scope[18]. Moreover, biological screening data for de novo pML libraries is rarely disclosed. Schreiber[19] and Myers[20] pioneering work in this area are notable exceptions, where libraries of up to 300 pMLs were constructed and screened in phenotypic and antimicrobial assays, respectively (Fig. 1B). Thus, the majority of de novo pML library design is driven by innovations in synthetic methods, such as oxidative ring expansion strategies[21,22] or organo- and metal catalysis strategies[23,24]. However, these methods-oriented approaches to pML library design tend to result in simplified pMLs that have yet to achieve natural product-like stereochemical and structural diversity.

To address these existing limitations, here we describe a structure-first approach to natural product-like library design (Fig. 1B). To achieve this goal, we focused on a region of chemical space populated by a small family of tetrahydrofuran (THF) containing pMLs exemplified by biselide A (Fig. 1A, see Supplementary Fig. 1 for additional examples). Importantly, each member of this constellation of natural products has a unique biological profile ranging from microtubule stabilizing[25] to antifungal[26], and as a consequence the THF pMLs have attracted considerable attention as leads for drug discovery and targets for total synthesis[27]. These combined features, coupled with our laboratories' expertise in the synthesis[28,29] of THF containing natural products and drugs[30], made the THF pMLs an attractive starting point for library design. Thus, we initiated a program designed to (i) map this region of chemical space using in silico assembled THF pMLs; (ii) score their functional group and volumetric overlap against their natural counterparts; (iii) exploit this information to populate relevant regions of THF pML chemical space through targeted synthesis; and (iv) evaluate the success of this approach through biological testing. Together, we term this approach the Targeted Sampling of Natural Product space (TSNaP) (Fig. 1C). The TSNaP approach builds on other strategies used to target natural product-relevant chemical space[31,32] and has similarities to pseudo-natural product design pioneered by Waldmann and co-workers[33–35]. In both TSNaP and pseudo-natural product design, compounds are synthesized using NP inspired synthetic fragments. However, unlike pseudo-natural product design, TSNaP prioritizes whole molecules rather than fragments based on their calculated structural similarity to a reference set of bioactive natural products. In principle, TSNaP can be applied to any sufficiently related family of known bioactives comprised of cyclic and acyclic fragments. Analogous to the 'habitable zone' that surrounds stars, TSNaP is designed to identify and exploit optimal regions of chemical space for the discovery of biologically relevant pMLs. This approach enables prioritization of molecules for targeted synthesis that are structurally related but sufficiently dissimilar to the natural product reference set to sample unexplored regions of natural product-relevant chemical space. Additionally, TSNaP overcomes limitations to molecular diversity found in the natural environment that are imposed by evolutionary trajectories, biosynthetic capabilities and building block accessibility. As highlighted here, this approach resulted in approximately 12% of the targeted library pMLs displaying activity in one or more of the assays evaluated, and produced compounds that closely resemble naturally occurring pMLs but exhibit different activities from the natural products that inspired them.

## Results and discussion

### Development of TSNaP for THF pML library design

The THF pML library design involved several stages. First, we examined the distinct classes of THF pML natural products[27] (fijianolides[25], biselides and haterumalides[26], phormidolides[36], and iriomoteolide-13a[37]) and identified 3 fragments that together comprised much of their shared molecular framework: Fragment 1: tetrahydrofuranol (9 distinct); Fragment 2: polyketide-like enoic acid (32 distinct); and Fragment 3: side chains (3 distinct: amide, thiazole, dihydropyran) (Fig. 2A). In addition, we retained the macrolactone ring to mirror structural features found in the original NPs and facilitate library construction. We then elaborated these fragments into synthetic building blocks that could be combined to create THF pMLs and largely retain the functional and peripheral groups found within the natural product reference set (Fig. 2B and Supplementary Fig. 1–3). The systematic combination of these fragments led to an in silico library of $9 \times 32 \times 3$ along with $2 \times 2$ potential stereoisomers, resulting in 3456 pMLs. Given the challenges associated with synthesizing a library of such structural complexity and size, a tool for prioritization of individual in silico pMLs for synthesis was required.

To reduce the number of in silico THF pMLs to a tractable synthetic problem (<100), we exploited the chemical information contained in the structures of the 18 natural products that comprised the reference set. First, all in silico generated and natural THF pMLs were subjected to a conformational search using Tinker (Fig. 2C)[38]. To ensure that all biologically relevant conformations of a given compound were considered, each conformer within 15 kcal/mol of the global minimum for that compound was retained based on studies by Chen and Foloppe who found this to be a relevant energy window for flexible macrocycles[39]. This process afforded a large collection of conformers for each in silico and natural compound. Second, a 3D structural similarity score ($C_s$) was calculated for in silico THF pMLs to provide a numerical assessment of their conformational and structural relatedness to the natural product reference set. To accomplish this, the volumetric and functional group overlap between each conformer of an in silico compound and each conformer of a particular natural product were compared using FastROCS[40,41] (OpenEye). For each in silico and natural THF pML pair, only the top scoring match against each conformation of the natural THF pML reference compound was retained and averaged (Fig. 2D), providing a collection of 18 unique scores for each of the 3456 in silico THF pMLs. The largest of these 18 scores (i.e., the best match with the natural products) was then used to define the 3D structural similarity ($C_s^{max}$, max value = 1) for each in silico compound (Fig. 2E). This analysis allowed us to prioritize compounds for synthesis that were sufficiently dissimilar to the natural products reference set ($C_s < 0.55$), but that contained at least one out of the six possible macrocycle-side chain analogues with relatively greater similarity ($C_s > 0.55$) to one or more of the natural products in the reference set. Critically, this process enabled prioritization of macrocycles that most closely resembled those found in the natural product reference set (maximum observed $C_s \sim 0.70$), which we hypothesized would help retain biological relevance. In most cases, only a single epimer of a single macrocycle-side chain combination (e.g. amide) mimics the natural product reference set ($C_s > 0.55$), while the other two (i.e. thiazole, dihydropyran) serve to diversify the backbone (and consequently tend to have lower $C_s$ scores), enabling a broader sampling of

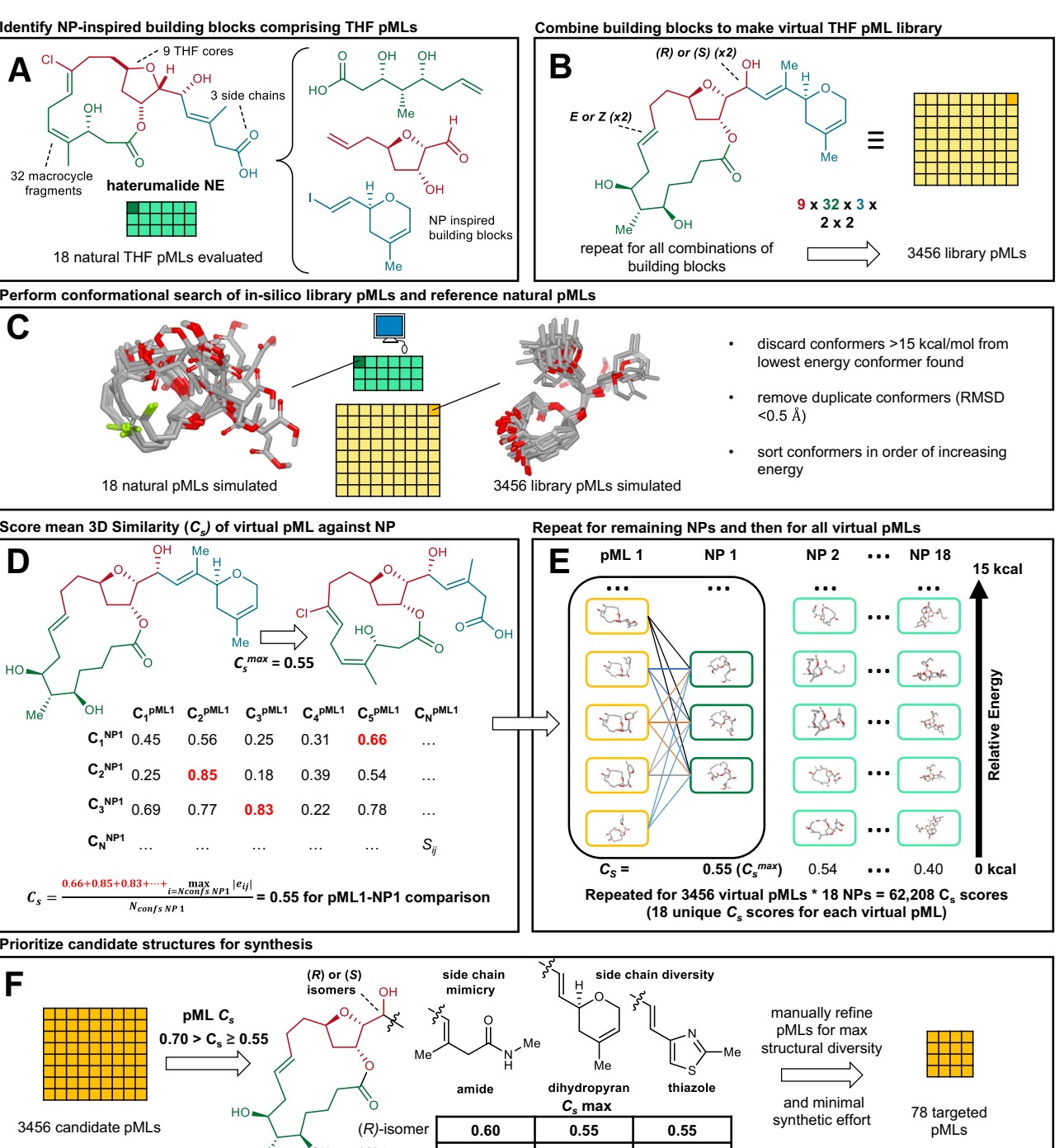

**Fig. 2 | Targeted sampling of natural product space (TSNaP). A** Identification of synthetically tractable NP-inspired building blocks (enoic acid (green), tetrahydrofuranol (red), side chains (blue)) that comprise the framework of natural THF pMLs. **B** Synthetic building blocks are systematically combined in silico to generate the entire virtual THF pML library of 3456 compounds. **C** All 3456 virtual THF pMLs and 18 reference natural products are subjected to a conformational search retaining all conformers within 15 kcal/mol of their respective minima. **D** Calculation of $C_s$ score between a single virtual THF pML and a single reference NP is achieved by (i) calculating all pairwise conformer-conformer volumetric and functional group overlap (TanimotoCombo score) (ii) retaining only the top scoring match against each conformation of the natural product and virtual THF pML

(iii) averaging these top scoring matches scores over the total number of conformations for that natural product to generate the $C_s$. **E** This process is then repeated against all 17 remaining reference NPs, and finally the entire process is carried out on all remaining 3455 virtual THF pMLs. This results in 18 unique $C_s$ scores for each virtual THF pML. **F** THF library pMLs were prioritized for synthesis if at least one epimer of one side chain-macrocycle analogue had a $C_s$ score between 0.55 and 0.70. In the indicated example, three out of the six candidate pMLs had a $C_s$ score between 0.55 and 0.70 and therefore all six compounds are considered targets for synthesis. Finally, the subset comprised of candidate structures was manually refined for structural diversity and minimal synthesis effort resulting in 78 THF pMLs prioritized for synthesis.

THF pML chemical space (Fig. 2F, Supplementary Fig. 4–6). Finally, to achieve maximum structural diversity with a minimum number of synthetic building blocks, we manually refined this virtual library to a final set of 78 target structures, which required a total of 16 unique building blocks.

## Synthesis of THF pML library

Considering the cumbersome nature of chiral polyketide synthesis, and the intimate dependency of macrocyclization chemistry on molecular structure and conformation, synthesis of a diverse and structurally complex library of THF pMLs presents real challenges. Thus, we invested considerable efforts in route scouting (to be reported elsewhere) and eventually identified the modular synthetic approach depicted in Fig. 3. We relied on three key coupling reactions using the readily produced building blocks tetrahydrofuranol **1a-e**[42], enoic acid and enoic aldehyde precursors **2a-h**, and vinyl iodides **3a-c**[43,44], and executed in order of esterification[45], ring closing metathesis[46] and Nozaki-Hiyama-Kishi (NHK) reaction[47]. As delineated for compounds **12a-c** and **12a-cAc**, production of each library member initiated with a Steglich esterification[45] of tetrahydrofuranol **8a**[42], which proceeded smoothly to furnish a diene precursor **4**. This material was then subjected to ring closing metathesis using Grubbs 2nd generation catalyst[48] to provide the desired macrocyclic lactone **5**. In most cases the macrocycle was produced in good yield, however, in several cases the inherent strain in the targeted macrocycle resulted predominantly in the formation the macrocyclic homodimers. In these problematic cases, formation of the desired macrocycle could be realized by increasing the reaction temperature (110 °C)[49], using shorter reaction times (5–15 min), and/or using the less sterically hindered Stewart-Grubbs 2nd generation catalyst[50]. Following macrocycle formation, deprotection using HF-pyridine provided the macrocyclic alcohol **6** in excellent yield. At this point, a sample of the macrocyclic alcohol was subjected to the appropriate conditions to expose all remaining peripheral alcohol groups and to provide **12d**, a standard macrocycle lacking a side chain.

Our initial attempts to oxidize the macrocyclic alcohol **6** and effect a subsequent NHK reaction were fraught with challenges that included low and irreproducible yields. After extensive investigation, we found that 1.3 equivalents of freshly purified Dess-Martin periodinane[51] consistently provided the corresponding macrocyclic tetrahydrofurfurals in good yield. These aldehydes were then subjected to NHK reactions[29,47] that introduce each of the three large side chains and proceeded in good yield over the two steps. In all cases a mixture of epimeric alcohols was produced with diastereomeric ratios ranging from 1:1 to >19:1 at the newly formed hydroxymethine stereocenter. Where possible, the diastereomeric alcohols were separated, however, in many cases these compounds proved to be inseparable by flash column chromatography (as indicated in Fig. 3). Finally, the peripheral alcohol functions were exposed using appropriate deprotection conditions, a process that generally proceeded in good yield and provided up to ~20 mg of material. Notably, in four cases (**10c** and **11a-c**) the deprotection step also resulted in translactonization with an alcohol function on the macrocycle, forming a 5- or 6-membered lactone. Despite this additional complexity, the majority (72) of the 78 targeted compounds **7**–**18a-c** (each a mixture of 2 diastereomers at point of side chain attachment) were successfully prepared following this strategy along with 12 macrocycles lacking a side chain (Fig. 4). In addition, a serendipitous oxidation product **7e** was produced during the oxidative cleavage of a PMB protecting group, resulting in a total of 85 unique THF pMLs. In an effort to modulate the polarity of THF pMLs and further mimic strategies used in nature (e.g. biselide A) to assist with passive diffusion of polar compounds[52,53], a small amount of each THF pML was peracetylated, providing a complimentary library of acetylated THF pMLs (e.g. **12a-dAc**). This resulted in a combined total of 170 unique library THF pMLs.

## Analysis of THF pML library

From a qualitative perspective, the completed library of THF pMLs show clear commonalities with the natural products reference set (Fig. 4). In particular, the peripheral groups on the macrocycles exemplify the polar and non-polar functional groups commonly encountered in THF pMLs (e.g., methyl branches, stereogenic alcohols and chlorides). Additionally, each large side chain incorporated in the synthetic THF pMLs mimics those encountered in naturally occurring pMLs[25,26,54]. To quantify the structural similarity between the natural products reference set and the synthetic THF pML library we grouped the $C_s$ scores for each of the 72 THF pMLs that were part of the prioritization phase by reference natural product subfamily (Fig. 4). This plot depicts the greatest $C_s$ score of all calculated THF pML candidates against each natural product subfamily (i.e. 5 scores for each THF pML), with the 72 synthesized molecules that were part of the design phase denoted by large spheres. This analysis revealed strong similarities to the haterumalide, biselide, and phormidolide families, but lower overlap with the fijianolide and iriomoteolide families. This result aligns well with the design philosophy for the pML library, given that both the stereochemistry at the embedded THFs and the macrocyclic ring sizes (14–16) most closely resembled the haterumalide, biselide, and phormidolide subfamilies of natural THF pMLs.

The 72 synthetic THF pMLs that were part of the prioritization phase (i.e. the non-acylated compounds with attached side chains) cover a wide range of $C_s$ values against the target natural products, ranging from 0.35 (low molecular overlap, low NP reference set likeness) to 0.64 (high overlap, high NP reference set likeness) (Fig. 4). Distributions between compound families varied, with most biselides-pML scores falling between 0.4 and 0.5 (low to moderate overlap), whereas most phormidolides-pML $C_s$ scores were between 0.5 and 0.6 (moderate to high overlap). Taken together, these qualitative and quantitative comparisons show that the library pMLs bear a characteristic resemblance to the natural products which inspired their development while also targeting previously uncharted chemical space.

## Biological evaluation of the pML library

To evaluate the biological activity profiles of THF pML library members we employed two whole-cell screening platforms; Cell Painting[55] and BioMAP[56]. Cell Painting is an image-based screening platform that uses five fluorescent stains to image different cellular components (nucleus, endoplasmic reticulum, nucleoli, cytoplasmic RNA, F-actin cytoskeleton, Golgi, plasma membrane, mitochondria). Segmentation of the resulting images and extraction of size and shape parameters for each cell in the image affords a numerical 'fingerprint' that describes the effect of compound treatment on cell development compared to untreated control wells. These fingerprints can then be clustered with fingerprints for control compounds with known modes of action to predict the biological functions of compound library members. As an independent measure of cell cytotoxicity, we also performed a standard MTT cytotoxicity assay on all library members. The BioMAP assay is an antimicrobial screening platform that includes 19 clinically relevant bacterial strains. Test compounds are screened as dilution series (16 × 2-fold dilutions) to give MIC activity profiles across the full panel. These activity profiles describe the spectrum of activity of each active compound and can be clustered to group compounds based on biological signature. As a complement to the BioMAP panel, the library was also screened against a matched panel of Gram-negative organisms (*Escherichia coli*, *Pseudomonas aeruginosa*, *Acinetobacter baumannii*), containing both wild-type and pore-overexpression/efflux-deficient strains[57,58] engineered to allow improved compound uptake and monitoring of target engagement within bacterial cells (see Supplementary information). The direct comparison of the wild-type and pore-overexpression/efflux deficient strains reflects the potential of individual molecules to penetrate through the bacterial outer-

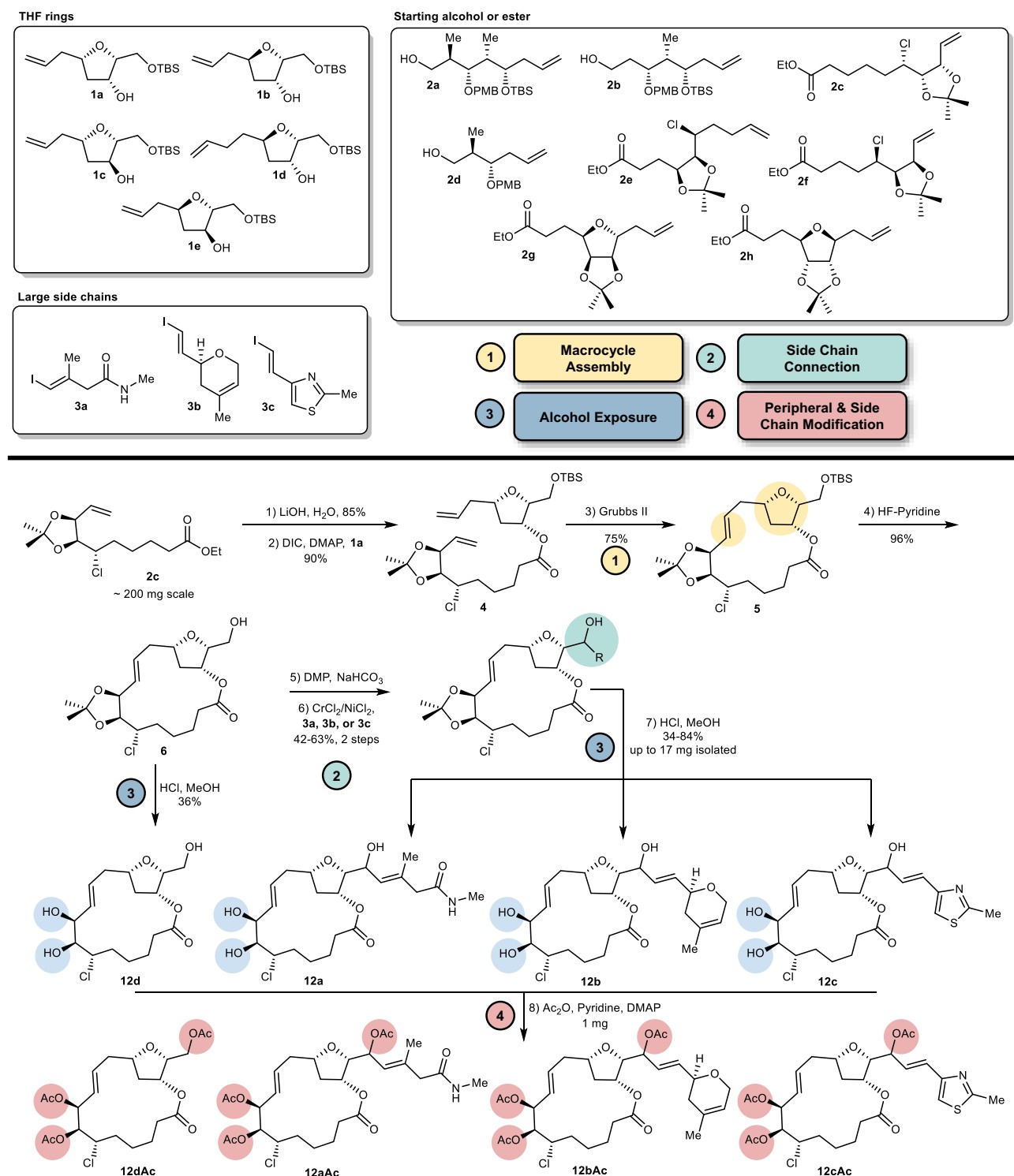

**Fig. 3 | Library building blocks and representative synthesis of a THF pML.** The 16 synthetic building blocks (THF ring, alcohol or ester, and side chains) used for generating a library of THF pMLs, and an example of the synthetic sequence used to produce **12a-d** and **12a-12aAc**. Key functional groups, and the corresponding steps to install or affix them, are as follows: macrocycle assembly to affix rigidifying groups (yellow), connection of the large side chain (teal), exposure of peripheral alcohols (blue), and peripheral group and side chain modification (red). Abbreviations: DIC = N,N′-diisopropylcarbodiimide, DMAP = 4-dimethylaminopyridine, Grubbs II = (1,3-Bis(2,4,6-trimethylphenyl)−2-imidazolidinylidene)dichloro(phenylmethylene)(tricyclohexylphosphine)ruthenium, DMP = Dess-Martin periodinane.

membrane and to assess properties to be modulated in order to improve the overall antibacterial activity.

All 170 library members were screened in both assay platforms, along with three natural product THF pML reference compounds from the original library design phase (phormidolide A, fijianolide A, and biselide A). Two of the natural products (fijianolide A and biselide A) exhibited potent cytotoxic activity in the MTT assay (80−160 nM; Fig. 5) and possessed Cell Painting profiles that clustered closely with microtubule-disrupting agents from the reference library, including Taxol and epothilone. However, none of the

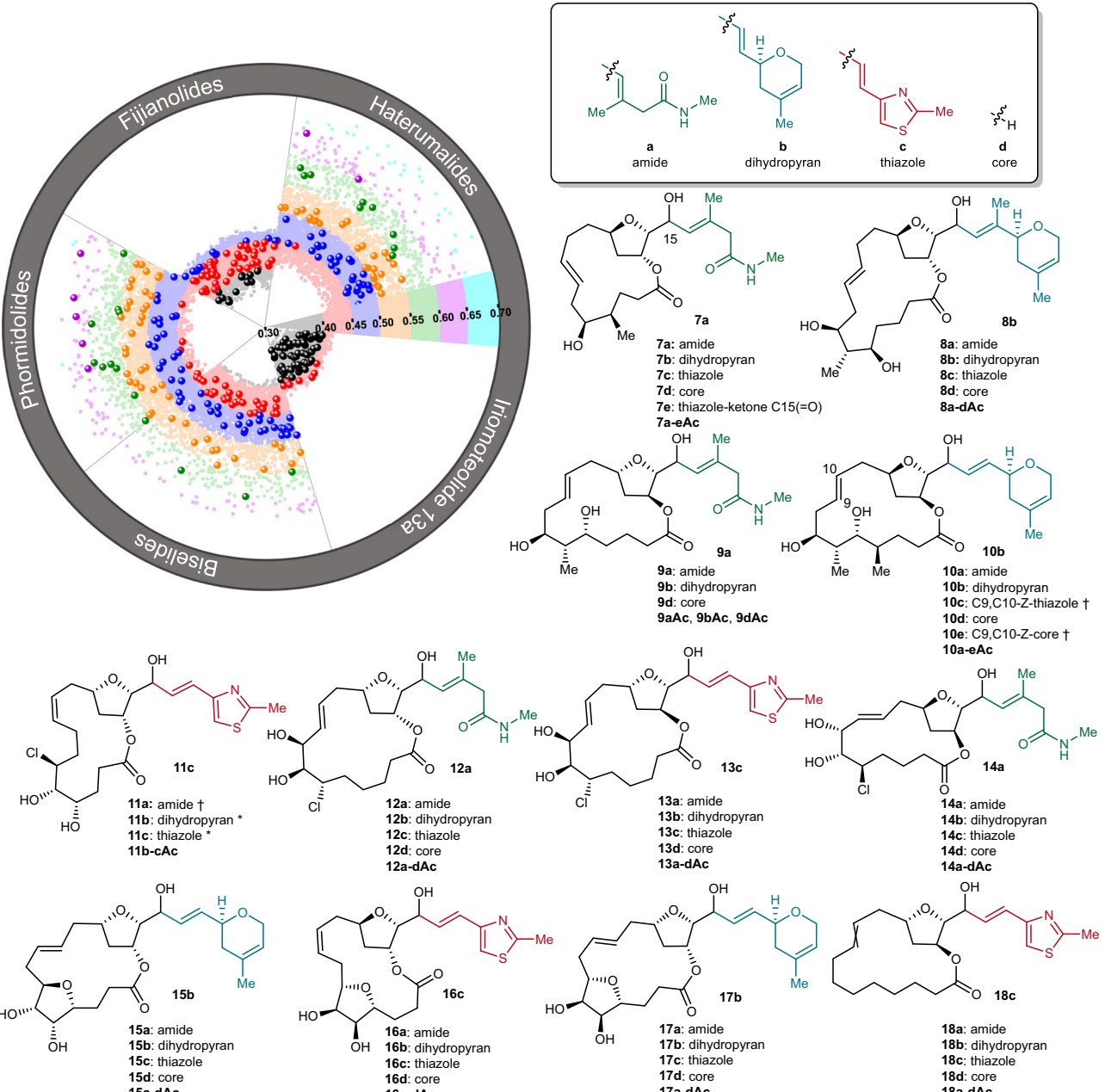

**Fig. 4 | The completed library of 170 THF pMLs.** In each case only a single side chain is depicted and the numbering (a-c) along with name (amide (green), dihydropyran (blue), thiazole (red)) refer to the attachment of one of the side chains **3a-c** depicted in Fig. 3. "Core" refers to the THF pML without an attached side chain. The addition of "Ac" to a compound number refers to the peracetate. Each compound was prepared as a mixture of diastereomeric alcohols at the point of side chain attachment. The circular plot depicts the largest $C_s$ scores (0.3 (grey) to 0.7 (cyan)) for synthetic (large spheres) and virtual (small spheres) THF pMLs from the design phase when compared to each subfamily of the naturally occurring THF pMLs. * isolated as a mixture of the translactonization (5 membered lactone) and macrolactone products. † isolated as the translactonization (5 or 6 membered lactone) products, which were not included in the total pML library count.

original natural products displayed any activity in the 19 strain BioMAP panel up to the highest tested concentration (32–64 µM). Screening of the synthetic macrolide library revealed the presence of five different bioactive classes (Fig. 5A), four of which possessed fundamentally different activities to those of the natural THF pMLs. Firstly, compounds (**16cAc, 9dAc, 7c-ketoneAc**, and **14cAc**) (blue group, Fig. 5A) displayed specific but moderate activity against three Gram-negative pathogens of the order Enterobacterales (*Klebsiella aerogenes, Salmonella enterica* and *Providencia alcalifaciens*). Interestingly, these compounds were not active against other Enterobacterales strains in the BioMAP panel. Notably, these compounds were also inactive in both the Cell Painting and MTT assays

up to the highest tested concentrations, indicating selectivity for prokaryotic over eukaryotic cells.

The second activity class (green group, Fig. 5A) included five library members (**18dAc, 18c, 18b, 18cAc**, and **10bAc**) with closely related structural features that were active against two Gram-positive and four Gram-negative pathogens (*Streptococcus pneumoniae, Ochrobactrum anthropi, Acinetobacter baumanii, Vibrio cholerae, Listeria ivanovii, Haemophilus influenzae*) with moderate potencies. Interestingly, many of the compounds in this group were also active against the pore overexpression strain of *E. coli* at 50 µM, but were not active against the other pore overexpression strains or the corresponding WT *E. coli* strain. Similarity between antibacterial profiles in

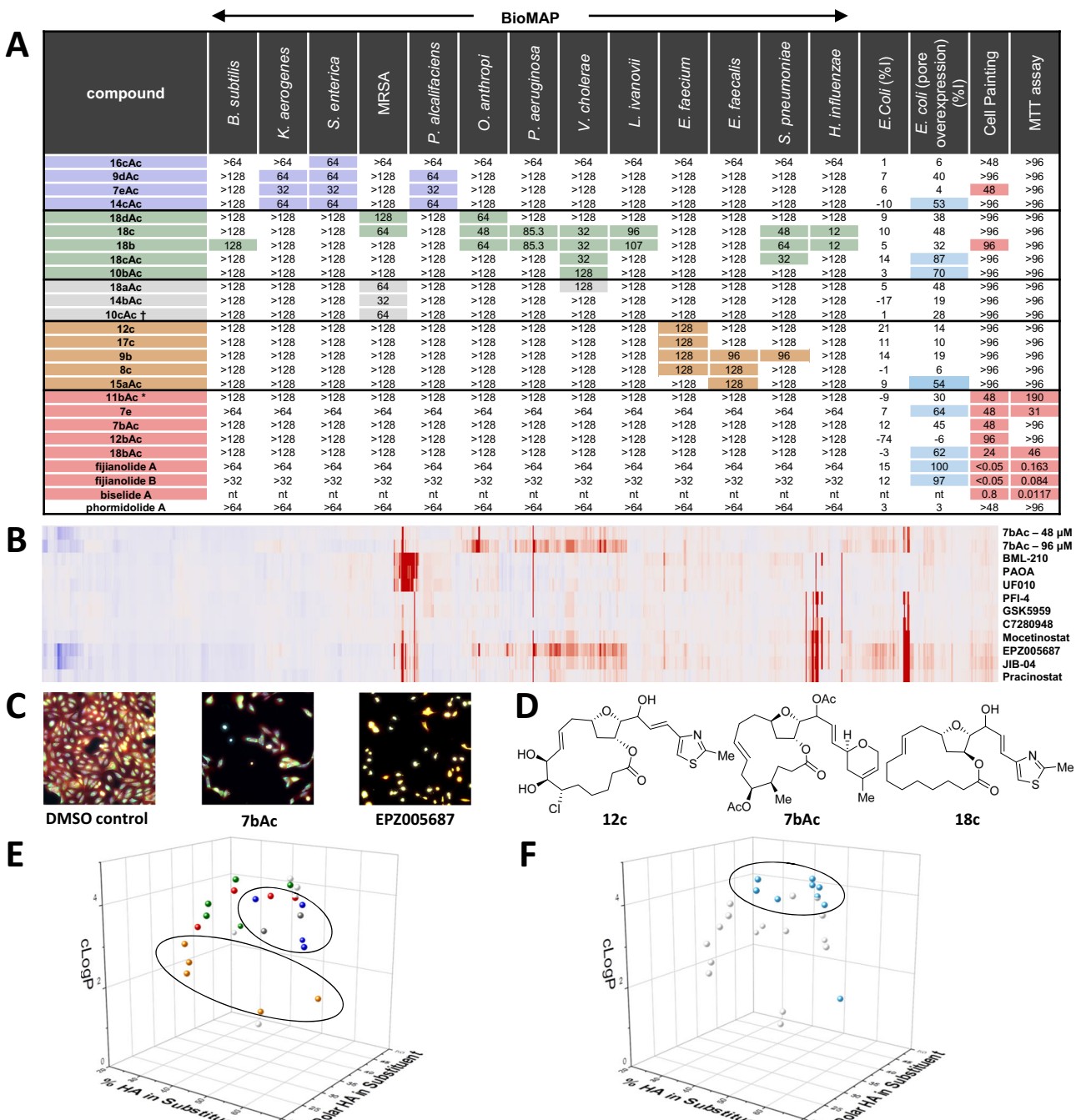

**Fig. 5 | Bioactivity data for active THF pMLs. A** Screening data arranged by compound activity class for 13 of the 19 BioMAP strains (gold, purple, green, and grey), the cell painting and MTT assays (red), and the matched wild-type *E. coli* and pore overexpression strain (teal). For the BioMAP, cell painting, and MTT assays the colored values indicate the lowest concentration (μM) at which a strong biological fingerprint was observed in these assays. For cell painting this is the minimum concentration where the square root of the sum of the squares of the fingerprint values is five times higher than the median value for the negative control wells. For the wild-type *E. coli* and pore overexpression strain, the teal values indicate the percentage inhibition during log-phase growth at 50 μM. * screened as a mixture with translactone (see Supplementary Information). "nt" denotes that the compound was not tested in that assay. † screened as a mixture containing the trans-lactonization product. **B** Cell painting fingerprints of THF pML **7bAc** and several histone acetylation, deacetylation, demethylation, and methylation inhibitors showing similar phenotypes. The colors within the fingerprints indicate that the

feature being measured is negative (blue), approximately the same (grey), or positive (red) relative to the DMSO control. **C** Cell painting images of a DMSO control, THF pML **7bAc**, and histone methylation inhibitor EPZ005687. EPZ005687 and **7bAc** show similar cell morphology and induce apoptotic cell death. **D** Structures of representative compounds for three of the activity classes. **E** 3-dimensional plot using % HA in substituents (*X*), % Polar HA in substituents (*Y*), and cLogP (*Z*) as parameters to describe all active THF pMLs (spheres). The active compounds are annotated by color matched activity class (gold, purple, green, grey, and red spheres). White spheres denote compounds active in a single assay that did not group with the five activity classes. **F** 3-dimensional plot using % HA in substituents (*X*), % Polar HA in substituents (*Y*), and cLogP (*Z*) as parameters to describe all active THF pMLs (spheres). This plot is identical to panel (**E**) but with THF pMLs active in the pore overexpression strain of *E. coli* highlighted as teal spheres, where as white spheres denote compounds that were inactive in this assay.

the WT BioMAP panel and selectivity between organisms in the pore overexpression panel suggests that these molecules may share a common mode of action, rather than merely possessing similar physicochemical properties (and therefore similar uptake profiles) for compounds with differing MOAs.

The third category (grey group, Fig. 5A) included two compounds (**18aAc, 14bAc**) with activities exclusively against MRSA. The fourth category (gold group, Fig. 5A) included five compounds (**12c, 17c, 9b, 8c, 15aAc**) with weak activity against either one or two closely related Lactobacillales strains (*Enterococcus faecalis* and *E. faecium*), but were inactive in all other assays. The final activity class (red group, Fig. 5A) contained five compounds (**11bAc, 7e, 7bAc, 12bAc, 18bAc**) which were active in the Cell Painting assay, yielding morphological profiles that were closely related and which grouped with reference compounds (Fig. 5B) resulting in apoptotic cell death (Fig. 5C). The reference compounds in this cluster are all involved in the disruption of histone function, including inhibitors of histone acetylation (PFI-4, GSK5959), deacetylation (BML-210, pracinostat, PAOA, UF010, mocetinostat), methylation (EPZ005687) and demethylation (JIB-04). Biselide A and fijianolides A and B have been shown to influence tubulin dynamics, which can be affected by histone function suggesting a possible overlapping role for these four macrolide scaffolds. Between the five activity classes, active compounds showed significant structural variation including diversity in (i) stereochemistry of the THF; (ii) macrocycle size and functionalities within the macrocycle; and (iii) side chain type (Fig. 5D). Overall, approximately 12% (21 out of the 170 synthesized THF pMLs) showed strong biological fingerprints in one or more of our panel of biological assays.

To interrogate whether compounds within the five bioactivity classes shared common structural or physicochemical features we calculated a suite of cheminformatics parameters (see Supplementary Tables 1, 2) and annotated the data by activity cluster. Several of the activity classes showed moderate to good clustering by percentage of heavy atoms (HA) in substituents, the fraction of polar HA in substituents, and cLogP (Fig. 5E). These parameters have been highlighted due to their importance in mediating the binding of macrocycles to protein targets[4], and serve as proxies for the relative size of substituents to the macrocycle backbone (% HA in substituents) and the degree of polarity of the substituents (% polar HA in substituents). The blue and gold compound classes formed distinct clusters by these parameters, with the gold compound class being characterized by a relatively low average cLogP (2.4) with approximately 28% of the HA in the substituents being polar. In contrast, members of the blue compound class have a greater cLogP (3.5) with a larger proportion of the HA in substituents (36%) being polar atoms. Additionally, compounds active in the pore over expression strain of *E. coli* clustered strongly by cLogP and the proportion of HA in their substituents (Fig. 5F). Clustering based on biological and physicochemical attributes implies that these compound groups may share mechanisms of action and could be the subject of future SAR studies. Together, these results reinforce our initial hypothesis that TSNaP produces biologically relevant chemical libraries with high chemical diversity which affect multiple biological targets.

Leveraging the structural information contained within a class of biologically active naturally occurring pMLs, we have developed a platform termed TSNaP that supports the creation of in silico libraries of natural product-like compounds and their prioritization for synthesis. By design, this approach targets molecules that are structurally related but sufficiently dissimilar to natural product reference sets to sample regions of natural product-relevant chemical space not otherwise accessible due to limitations on biosynthetic capabilities. Using this approach, we report the modular synthesis of a THF pML library with $C_s$ values against the natural product references ranging from moderate to high natural product reference set-likeness. Finally, we demonstrate that TSNaP results in chemical matter with hit rates

that exceed those typically encountered in small molecule libraries, and biological profiles that are distinct from the natural products used for parameterization. Thus, TSNaP represents an enabling tool that compliments existing strategies to map chemical space and should help guide the design of improved chemical libraries for drug discovery purposes.

## Methods

### Generation and parametrization of in-silico library pMLs and Natural THF Macrolides

Structures for each of the natural THF macrolides (Supplementary Fig. 1) were acquired from a Reaxsys® search. SMILES for the building blocks side chains (e.g. SC_1), macrocycle body (e.g. MC_1), and THFs (e.g. THF_1) of the library pMLs were combined to generate SMILES for each of $32 \times 9 \times 3 \times 2 \times 2$ library pMLs allowing for both *E/Z* and *(R)/(S)* configurations at the ring closing metathesis and Nozaki-Hiyama-Kishi centres, respectively (see Supplementary Fig. 2, 3). SMILES for the 3456 pMLs and the 18 natural products (or truncated fragments of natural products, see Supplementary Fig. 1), were converted to initial geometry files with RDKit (version 2017.09.01)[59] and parametrized for the MMFF94[60] forcefield with sdf2tinkerxyz[61].

### Conformational Sampling of in-silico library pMLs and Natural THF macrolides

The molecules described in the preceding section were simulated with the Tinker molecular modelling package[38] (version 8.2). Each molecule was simulated using the generalized Born Still-approximation solvation model[62] and a distance dependent dielectric constant of 78.3. Each molecule was subjected to 2500 individual molecular dynamics simulations of 1 picosecond at 1000 K using the Berendsen temperature coupling bath, with a time step of 1 femtosecond. The first iteration started from the geometries described in the preceding section, and the resultant geometry was then energy minimized using a Newton-Raphson minimization algorithm to an atom root-mean-square gradient of 0.01 kcal/mol/Å. Subsequent MD steps used the energy minimized geometry from the previous minimization step until a total of 2500 iterations were reached. Throughout the course of the simulations, stereochemistry was enforced, and a soft harmonic constraint ($k = 0.005-0.01\,kcal/deg^2$) was applied to the torsion angles of all double bonds when they rotated beyond 45° from their ideal geometry (i.e. *E* or *Z*).

After completion of the conformational sampling protocol the resultant 2500 conformations were sorted from lowest to highest energy. Conformations with an energy greater than 15.0 kcal/mol relative to the lowest energy conformation identified for each individual molecule were discarded. Finally, all pairs of conformers were overlaid, and their pair-wise atom RMS displacements were calculated. If the atom RMS was <0.50 Å between any pair of conformers, the higher energy conformer was discarded.

### Conformer-conformer scoring protocol and calculation of $C_s$

FastROCS TK (OpenEye toolkits version 2017.10.1) was used to calculate all TanimotoCombo (i.e. volumetric and functional group overlap) scores between the library pML ($N = 3456$) conformers and the natural product reference set ($N = 18$) conformers. The default color forcefield setting (implicit Mills Dean) was used for all color overlap calculations. Only the top scoring TanimotoCombo scores (i.e. values in red in Fig. 2D) were output and were subsequently averaged using either Excel or python version 3.6. All overlap calculations were carried out on NVIDIA Tesla P100 or V100 GPUs.

### Synthetic methods and materials

All reactions described were performed under an atmosphere of dry nitrogen using oven or flame dried glassware unless otherwise specified. Flash chromatography was carried out with 230–400 mesh silica

gel (Silicycle, SiliaFlash® P60, Merck). C2 modified/deactivated silica gel was prepared according to the procedure reported in the literature[63]. 10% (w/w) silver nitrate impregnated silica gel was prepared by slurrying 10 g AgNO₃ with silica gel (100 g) in 60 mL H₂O. The slurry was dried in an oven at 150 °C for several hours until dry. Concentration and removal of trace solvents was done via a Büchi rotary evaporator using a dry ice/acetone condenser and vacuum applied from a Büchi V-500 pump. Microwave reactions were performed in a CEM discovery LabMate microwave reactor. All reagents and starting materials were purchased from Sigma Aldrich, Alfa Aesar, TCI America, Oakwood Chemical, Combi-Blocks, Carbosynth, or AK Scientific and were used as received without further purification. All solvents were purchased from Sigma Aldrich, Fisher or ACP and used without further purification unless otherwise specified. Dichloromethane ($CH_2Cl_2$) was freshly distilled from calcium hydride. THF was freshly distilled from sodium metal/benzophenone. Anhydrous DMSO in SureSeal™ bottles was purchased from Sigma-Aldrich and used without further purification unless otherwise specified. All other organic solvents were dried over activated 3 Å molecular sieves (20% m/v). Cold temperatures were maintained by use of the following conditions: 0 °C, ice/water bath; −40 °C, acetonitrile-dry ice bath; −78 °C, acetone-dry ice bath. For all other temperatures, a bath consisting of the appropriate mixture of MeOH/H₂O and dry ice was used.

## General synthetic procedures

**General Procedure A. Acetylation of pMLs.** To a mixture of pyridine (5 mL) and acetic anhydride (5 mL) was added DMAP (0.5 mg, 0.004 mmol). To a vial containing the macrocyclic alcohol was added the mixture of pyridine/acetic anhydride/DMAP (0.5 mL). After stirring for 1–2 h, the mixture was concentrated under reduced pressure (<1 torr) until dry. The residue was dissolved in EtOAc and run through a short plug of silica gel eluting with further EtOAc (ca. 5–10 mL), and the eluted liquid was blown down to dryness with air and purified via flash column chromatography.

**General Procedure B. Ring-closing metathesis.** To a warm (60 °C), solution of diene (1 eq.) in dry deoxygenated toluene (ca. 2.5 mM in diene) with N₂ sparging was added a solution of Grubbs II catalyst (0.20 eq.) in a small volume of toluene (ca. 1–5 mL). The reaction mixture was allowed to stir under continuous N₂ until consumption of starting material as monitored by TLC analysis. Subsequently, the reaction mixture was cooled to room temperature and quenched by addition of potassium 2-isocyanoacetate (excess) in methanol (ca. 1–10 mL). After stirring for 1 h at room temperature, the mixture was concentrated to dryness, filtered, and solvent was removed in vacuo. The crude product was purified via flash column chromatography with the appropriate eluent to provide the pure protected macrocyclic cores. Precise reaction conditions can be found for each ring-closing metathesis in the Supplementary information.

**General Procedure C. Nozaki-Hiyama-Kishi (NHK) reaction.** Anhydrous CrCl₂ was doped with 1% NiCl₂ (w/w) in a glove box and thoroughly mixed by hand for a minimum of 30 min. to create a stock mixture of CrCl₂/NiCl₂. A custom piece of glassware was designed and built (see Supplementary Fig. 26) to perform the NHK reactions, consisting of a solid addition arm fitting into a 2-necked vial via a ground glass joint. The apparatus is charged with a small magnetic stirring bar and the solid addition arm is rapidly charged with the previously prepared mixture of CrCl₂/NiCl₂ (-10–15 eq. CrCl₂, -0.10–0.15 eq. NiCl₂). The apparatus is then promptly assembled, fitted with a rubber septum, and evacuated/backfilled with a vacuum pump and dry nitrogen gas. Subsequently, the macrocyclic aldehyde precursors (1.0 eq.) are dissolved in a small volume of dry and oxygen free DMSO (-0.02–0.1 M in macrocyclic aldehyde), syringed into the apparatus, and stirring initiated. The solid addition arm is then slowly rotated to allow portion-wise addition of the CrCl₂/NiCl₂ to the reaction mixture. Next, the appropriate vinyl-iodide (**3a-3c**, -1.5–3 eq.) is added either as a neat oil or as a concentrated solution in DMSO and the reaction mixture is allowed to stir overnight at room temperature (typically 12–24 h). After this time, the reaction mixtures were pipetted onto a mixture of water/brine and diethyl ether. The aqueous layer was repeatedly extracted with diethyl ether until no more product remained in the organic extract layer. The organic layers were dried (MgSO₄), filtered, and solvent was removed in vacuo to give the crude product. The crude product was purified via flash column chromatography using the appropriate eluent to give the pML product. Precise reaction conditions can be found for each NHK reaction in the Supplementary information.

## NMR spectroscopy methods and materials

Nuclear magnetic resonance (NMR) spectra were recorded using chloroform-d (CDCl₃), dichloromethane-d2 (CD₂Cl₂), methanol-d4 (CD₃OD), DMSO-d6 ((CD₃)₂S = O), benzene-d6 (C₆D₆) or acetonitrile-d3 (CD₃CN). Signal positions (δ) are given in parts per million relative to tetramethylsilane (δ 0.00) and were measured relative to the signal of the residual non-deuterated solvent ¹H NMR: CDCl₃: δ 7.26; CD₂Cl₂: δ 5.32; CD₃OD: δ 3.31; (CD₃)₂S = O: δ 2.50; C₆D₆: δ 7.16; CD₃CN: δ 1.94 ¹³C NMR: CDCl₃: δ 77.16; CD₂Cl₂: δ 53.84; CD₃OD: δ 49.0; (CD₃)₂S = O: δ 29.84; C₆D₆: δ 128.06; CD₃CN: δ 1.32. Coupling constants ($J$ values) are given in Hertz (Hz) and are reported to the nearest 0.1 Hz. ¹H NMR spectral data are tabulated in the order: multiplicity (s, singlet; d, doublet; t, triplet; q, quartet; m, multiplet; br, broad), coupling constants, number of protons. NMR spectra were recorded on a Bruker Avance 600 equipped with a QNP or TCI cryoprobe (600 MHz), Bruker 500 (500 MHz), or Bruker 400 (400 MHz). Assignments of ¹H and ¹³C NMR chemical shifts are based on analysis of ¹H-¹H COSY, HSQC, HMBC, and 2D NOESY spectra, where applicable. Raw NMR data was processed in MestreNova (version 14.1). ¹H NMR data was Fourier transformed with no apodization function, and the resultant NMR spectrum was phase corrected and baseline corrected with either a Whittaker smoother or ablative algorithm. ¹³C NMR data was Fourier transformed with a 0.5–1.0 Hz exponential apodization function, and the resultant NMR spectrum was phase corrected and baseline corrected with either a Whittaker smoother or polynomial function.

## Polarimetry

Optical rotation measurements were recorded on a Perkin Elmer 341 polarimeter at 589 nm and 20 °C in the indicated solvent at the indicated concentration denoted as 'c' and having units of g/100 mL. All measurements were recorded in a 1 mL cell with a path length of 1 dm. Prior to all measurements a blank solvent sample was run and zeroed for each sample with the solvent blank being the same solvent used for the dissolution of the sample. The cell was oriented in the exact same manner for both solvent blank runs and sample measurements.

## Infrared spectroscopy

Infrared (IR) spectra were recorded on a Nexus 670 Fourier transform spectrometer on solvent free samples (neat). Only selected, characteristic absorption data are provided for each compound. A multi-point baseline correction and smoothing algorithm (30–50 points) was used prior to reporting the selected peaks for each compound.

## Mass spectroscopy

High resolution mass spectrometry was performed on either Agilent 6210 TOF LC/MS using ESI-MS, Waters SYNAPT UPLC-ESI-qTOF, or Waters RDa ESI-TOF. High performance liquid chromatography (HPLC) was performed on an Agilent 1200 Series equipped with a variable wavelength UVVis detector (λ = 220 nm) using Phenomenex Kinetix XB-C18 column (4.6 × 250 mm, 5 μm). Mass spectra were processed in Mestrenova (version 14.1).

## BioMAP. Antimicrobial screening protocol

The BioMAP assay was performed using a modified version of the protocol outlined in the original publication[56], as used in several recent studies[64–67]. Antimicrobial susceptibility testing for pMLs **7–18 and 7Ac-18Ac** against a 19-member bacterial target panel (Supplementary Table 5) were performed using a 384-well high-throughput screening protocol adapted from the broth microdilution methods developed by the Clinical and Laboratory Standards Institute (CLSI). Bacterial test strains were individually grown on fresh Nutrient Broth (NB, ATCC Medium 3) agar, Tryptic Soy Broth (TSB, ATCC Medium 18) agar or Brain Heart Infusion (BHI, ATCC Medium 44) agar, respectively, as recommended by the American Type Culture Collection (ATCC) from which test strains were obtained. Each strain was streaked on solid agar and individual colonies used to inoculate 3 mL of sterile NB, TSB or BHI media depending on test organism. These small-scale liquid cultures were grown overnight with shaking (200 RPM; 37 °C). *Listeria ivanovii* (ATCC BAA-139) and *Streptococcus pneumoniae* (ATCC 49619) were incubated overnight but not shaken (37 °C; 5% $CO_2$). Saturated overnight cultures were diluted in their respective media according to turbidity to achieve approximately $5 \times 10^5$ CFU/mL of final inoculum density and dispensed into sterile clear polystyrene 384-well microplates (Thermo Scientific™ 265202) with a final screening volume of 30 µL. *L. ivanovii* was diluted with and grown in Haemophilus Test Medium (HTM; ATCC Medium 2167).

Solutions of test compounds and antibiotic controls were prepared as 1:1 dilution series in DMSO (16 x two-fold dilutions for each compound, final testing concentrations 128 µM to 3.91 nM). Compound plates were pinned into each assay plate (200 nL) using a high-throughput pinning robot (Tecan Freedom EVO 100; V&P Scientific Pin-tool). In each 384-well plate; lane 1 was reserved for DMSO vehicle and culture medium; lane 2 reserved for DMSO vehicle, culture medium, and target bacteria; lanes 23 and 24 reserved for antibiotic controls, DMSO vehicle, culture medium, and target bacteria. After compound pinning, assay plates were read as $t_0$ at $OD_{600}$ using an automated plate reader (BioTek Synergy Neo2), covered with a plastic lid and placed in a humidity-controlled incubator at 37 °C for 18–20 h before $OD_{600}$ readings were taken at $t_{20}$. *L. ivanovii* and *S. pneumoniae* were incubated in a separate incubator (37 °C; 5% $CO_2$). Resulting growth curves for each dilution series were used to determine the MIC values for all test compounds following standard procedures.

## Cell painting protocol

A dilution series (96 µM–1.5 nM, 16-point x 2-fold dilutions) of each pML **7–18** and **7Ac-18Ac** was applied to U2OS (ATCC HTB-96) human osteosarcoma cells and incubated overnight. The cells were then treated with a standard set of five fluorescent stains, according to the cell painting protocol described by Bray et al.[55]. Images were obtained via an ImageXpress XLS microscope (20X objective; Molecular Devices, CA, USA). Cell Profiler[68] was utilized to extract approximately 1000 features, including fluorescence intensity, cell number, granularity, and texture features, among others, from the obtained images. Raw features obtained from Cell Profiler were normalized according to Bray et al.[55]. Subsequently, biological activity profiles (fingerprints) were generated for Fijianolides A & B as well as each synthetic pML that displayed activity in this assay (**7e, 7eAc, 18b, 18bAc, 11bAc, 12bAc**; Supplementary Fig. S18A–S25A).

Activity scores were generated by first filtering the raw fingerprint to remove columns containing values greater that 3 standard deviations above the median value for each concentration. Next, the square root of the sum of the squares for all fingerprint values was calculated and plotted vs. concentration (Supplementary Fig. S18B–S25B). Finally, Activity Scores were calculated for negative control wells (DMSO vehicle only) and active concentrations defined as those concentrations where the activity score was more than 5 times above the mean activity score for negative control wells (DMSO vehicle only).

To identify potential modes of action of the pMLs **7–18** and **7Ac-18Ac** hierarchical clustering was performed on fingerprints from cells treated with the pMLs **7–18** and **7Ac-18Ac** and fingerprints from a reference library of 4400 compounds of known function (Targetmol, L4000). Orange Data Mining software[69] was used to perform hierarchical clustering, as well as to visualize hierarchical clustering dendrograms and associated heatmaps.

## MTT assay protocol

The U2OS (HTB-96, American Type Culture Collection, ATCC) osteosarcoma cell line was seeded into 384 well plates at a density of 2400 cells per well in 40 µl of McCoy's 5 A media (30–2007, ATCC). The cells were then incubated for 24 h in a 37 °C incubator. Subsequently, cells were treated with a dilution series (96 µM–1.5 nM, 16-point x 2-fold dilutions) of pMLs **7–18** and **7Ac-18Ac**. The cells were then incubated for a further 24 h. The following day the media was replaced with 40 µl of fresh media and cells were treated with 5 µl of MTT (4 mg/ml stock solution, M6494, ThermoFisher Scientific), centrifuged at 125× *g* for 30 s, and incubated at 37°C for 4 h. Post incubation, all but 10 µl of the media-MTT solution was removed and 20 µl of DMSO was added to each well. Cells were centrifuged at 300× *g* for 3 min, shaken at 800 rpm for 15 min using a Bioshake (Q Instruments) and incubated for 10 min at 37 °C. Subsequently, the cells were shaken at 800 rpm for a further 15 min, following which absorbance was read at 540 nm via plate reader (Synergy Neo2, Agilent).

The controls for this assay included 8 MTT only wells (which contained MTT, DMSO and cell culture media) and 16 'vehicle,' or DMSO wells (which contained cells, media, DMSO and MTT). Raw absorbance values were adjusted by subtracting the averaged background absorbance from the MTT only wells according to Larsson et al.[70]. Percent viability was obtained by dividing the adjusted absorbance values for the treatments by the averaged adjusted absorbance values for the DMSO controls[70]. GraphPad (version 8.4.3) was utilized to generate dose response curves and determine IC50 values.

## Antibacterial susceptibility single-point screening in pore-overexpressing/efflux-deficient bacterial strains and their wild-type controls

Purpose and principle: single point concentration assay to determine antibacterial potency (%inh). Read-out at λ = 600 nm to measure absorbance as a measurement of bacterial cell growth.

Antimicrobial susceptibility testing of wild-type and pore-over-expressing/efflux-deficient bacterial strains was performed as previously described by Krishnamoorthy et. al.[57]. Compounds were mixed with 50 ul of bacterial inoculum (-105 cells/ml; 0.2% arabinose to induce pore-overexpression) in Greiner 384-well plates (#781096). Sealed plates were incubated for 16 h at 37 °C and absorbance values (600 nm) were recorded every 20 min by a Biotek Synergy Neo.

## Reporting summary

Further information on research design is available in the Nature Portfolio Reporting Summary linked to this article.

# Data availability

Raw NMR data files for all final pML library compounds have been deposited in the Zenodo database under accession code https://doi.org/10.5281/zenodo.10576116. Raw data for the cell painting fingerprint generation and activity cutoff plots have been deposited in the Zenodo database under accession code https://doi.org/10.5281/zenodo.10576116. All conformers of the natural products reference set compounds and the associated maximum TanimotoCombo scores used to calculate the $C_s$ among the natural products have been deposited in the Zenodo database under accession code https://doi.org/10.5281/zenodo.10576116. All conformers of the in silico pMLs and natural products reference set compounds and the associated

maximum TanimotoCombo scores used to calculate the $C_s$ between natural products and pMLs have been deposited in the Zenodo database under accession code https://doi.org/10.5281/zenodo.10576116. Experimental procedures, chemical characterization data and processed ¹H and ¹³C NMR spectra are provided in the Supplementary Information. Source data are provided with this paper.

## Code availability
Scripts for calculating the TanimotoCombo scores were based on modified code from OpenEye scientific (https://docs.eyesopen.com/toolkits/python/fastrocstk/examples_summary.html) using example BestShapeOverlay.py. Copyright terms from OpenEye scientific do not permit the reproduction of the modified code base.

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

## Acknowledgements

Financial support from the Natural Sciences and Engineering Research Council (NSERC) of Canada was received by R.B. (Discovery Grant: 2019-06368), R.G.L. (Discovery Grant: 2021-02979), D.M.W. (CGS-M & PGS-D) and D.J.D (CGS-D). The authors thank OpenEye Scientific Software for providing a free academic license to their software which enabled the 3D overlap calculations and similarity comparisons. The developers of RDKit, sdf2tinkerxyz, and Tinker are also acknowledged. The Digital Research Alliance of Canada (formerly Compute Canada) is thanked for providing access to the high-performance computing resources that enabled this study. We thank Venugopal Rao Challa (SFU) for providing an authentic sample of biselide A, Tyler Johnson (DUC), Joseph Morris (DUC), Phil Crews (UCSC) and Erin McCauley (CSUDH) for providing an authentic sample of fijianolide B for chemical conversion to fijianolide A, and William Gerwick (UCSD) and Kelsey Alexander (UCSD) for providing an authentic sample of phormidolide A. The Zgurskaya laboratory (University of Oklahoma) is acknowledged for providing wildtype, pore-overexpression, and efflux deficient bacterial strains.

## Author contributions

R.B., R.G.L. and D.M.W. designed the study, R.B., D.M.W. and D.J.D. developed the synthetic plans and D.M.W. and D.J.D. optimized and executed the synthesis of all new compounds. D.M.W. optimized, performed, and interpreted the results for the conformational sampling, 3D scoring calculations, and cheminformatics analysis. D.Y.L., S.K., A.H. and C.B. carried out the biological screening, R.G.L. interpreted the biological screening data. D.M.W, D.J.D. and D.Y.L prepared the manuscript figures and R.B., R.G.L., D.M.W, and D.J.D. prepared the manuscript text. D.M.W, D.J.D, D.Y.L, S.K, A.H, and C.B. prepared their respective sections of the methods and supplementary information.

## Competing interests

The authors declare no competing interests.

## Additional information

**Supplementary information** The online version contains
Supplementary Material available at

Roger G. Linington or Robert A. Britton.

peer review file is available.

