## [Peer Review File · Nature Communications]

REVIEWER COMMENTS

Reviewer #1 (Remarks to the Author):

Britton et al. describe targeted sampling of natural product space as a new molecular design concept to explore focused yet prevalidated areas of biologically relevant chemical space. Highly complex, yet highly biologically relevant tetrahydrofuran-containing polyketide macrolides were chosen as starting points and were deconstructed into fragments and reassembled with all possible combinations in silico to afford >3000 virtual molecules. Conformational and functional group overlap was scored and virtual compounds that had the lowest similarity relative to the guiding NPs were prioritized for synthesis. A convergent synthetic approach utilizing the groups expertise in macrolide synthesis was used to access 170 compounds (including acetylated derivatives) from a small set of unique building blocks. Biological performance of the collection was evaluated by BioMAP and morphological profiling, revealing substantially higher hit rates than typical screening collections and adding support for the design principle as a means to efficiently access novel biologically active chemical matter.

Overall, this is a nice piece of work that has rationally and efficiently given access to novel structures that lie in an area of biologically-rich chemical space. The conclusions are consistent with the data presented, the manuscript is well written, and the methods and compounds are well characterized in the supporting information. Of note, it is particularly impressive that the designed and synthesized compounds retain the complexity of the parent natural products, which is not usually the case for collections inspired by polyketide macrolides.

This is an important contribution to the field, and I would recommend this work for publication in Nature Communications after addressing these minor points:

1. The introduction mentions that macrocycles typically show poor adherence to conventional rules for drug-likeness but yet are able to enter cells and be successful drugs. Some explanations are given how they can overcome these obstacles, but it would be beneficial elaborate on the possibility of solute carrier proteins as an alternative to passive diffusion (Nat. Rev. Drug Discov. 2008, 7, 205-220). This is especially relevant for this study as the compounds tested are highly natural product-like and may therefore have recognition elements for transport proteins.
2. The proposed TSNaP approach has several similar features to the pseudo-natural product design concept (J. Am. Chem. Soc. 2022, 144, 3314). Both concepts deconstruct natural products into natural product fragments and recombine them in different arrangements. However, the pseudo-natural product concept provides cyclic fragments whereas the TSNaP approach can provide both cyclic and acyclic fragments. TSNaP may be complimentary to pseudo-natural product design and, as demonstrated, may be much more useful when employing macrocyclic natural products as starting points. A brief comparison between the two methods should be included in the introduction.

3. Please provide a general range of diastereoselectivities for the Nozaki-Hiyama-Kishi reactions in the main text.
4. Figure 4 - I would suggest using R groups for the side chains to make the structures more representative.
5. Figure 4 – The dots in the outer region (0.65-0.7) are colored orange but should be colored cyan to match the legend.
6. Figure 4 – The collection was prepared as a mixture of diastereomers from the NHK reaction. Were both diastereomers considered for the Cs scores or just one diastereomer? Please clarify this in the text and/or in the caption of Figure 4.
7. The caption in Figure 5 states ‘indicate the lowest concentration (μM) at which a strong biological fingerprint was observed’. In the context of the cell painting assay, what does this mean? Is it purely qualitative or is there a quantitative threshold? Please explain this in the caption.
8. Figure 5b – Could you provide comparative CPA fingerprints to show that 7bAc and BML-210 are similar (or to other apoptosis-inducing compounds)?
9. Page 14 – ‘(low overlap, moderate NP-likeness)’, ‘(high overlap, closer similarity to NP scaffolds)’, and page 20 ‘high natural product-likeness’. I believe the intentions are to compare to reference NPs not NPs in general. Additionally, Cs is not comparing scaffolds but 3D-structural overlap. These should be rephrased to resemble more closely what the comparisons are.

Reviewer #2 (Remarks to the Author):

Linnington, Britton and coworkers report a fragment-based approach to the design of natural product-like libraries. The approach is guided by insights into key structural features of polyketide macrolide natural products and deconstruction into synthetically-accessible fragments, which then can be recombined into natural product-like macrocycles. There are several attractive features of this work, including the macrolide target class, computational approach to library design, descriptions of reaction

troubleshooting, size of the library, and demonstration of biological activity. However, the manuscript is also unclear in some places and further makes several expansive claims that are not sufficiently supported by the data. As such, while the work is, in principle, appropriate for publication in Nature Communications, numerous significant revisions are required prior to publication.

Key issues:

- Scholarship (p6) - Extensive previous work by other groups, particularly Waldmann, on natural product fragment-based libraries is not cited. The senior authors should do a comprehensive literature search, cite the most relevant prior art in the field, and revise the introduction to place the current work in proper context.
- p9 - The approach considers all conformers within 15 kcal/mol of the global minimum for each compound. This is a massive range, the rationale for which is not explained sufficiently in the text. Are such high-energy conformations even accessible in the context of the binding pose with a macromolecular target?
- p9 - "This analysis allowed us to prioritize compounds for synthesis that were sufficiently dissimilar to the natural products training set ($C_s < 0.55$), but that contained at least one macrocycle side-chain analogue with relatively greater similarity ($C_s > 0.55$)."
- This statement is confusing as it implies that the side chains were analyzed separately from the complete structures. Only after reading this and Figure 1F several times did it become clear that the authors are referring to sets comprised of a single scaffold with the 3 different side chains. The $C_s < 0.55$ criteria is also not shown in Figure 1F nor stated in the caption. This presentation needs to be revised.
- Figure 3 - From the total library of 3456 possible structures, the authors selected a subset of building blocks for synthetic feasibility to target 78 library members. However, the 5 x 8 x 3 building blocks selected provides a maximum of 120 combinatorial structures. The authors should comment on why the other 42 structures were not selected for synthesis.
- p13 - The production of diastereomeric mixtures detracts from the elegance of the library design and synthesis, especially as the compounds are screened as mixtures that may have independent biological activities. Did the authors investigate any asymmetric or diastereoselective alternatives to the NHK coupling? What is the dr? Are the diastereomers separable?

- p13 - Acetylation of the library increases the number of compounds produced, but these compounds will certainly be metabolically unstable in vivo (in animals). The authors cite literature on potential utility in vitro (cell-based assays), but do not note to this major downstream liability.

- Figure 4 - There appear to be 48 non-acetylated compounds shown, with 47 acetylated compounds, for a total of 95 structures. If diastereomers are counted separately, there would be $96 + 94 = 190$ structures. Yet, the library size is stated as 170 compounds; this needs to be clarified.

- p16 - The use of Gram-negative pore-overexpression strains is mentioned, but the rationale for this is not described in sufficient detail, there are no references cited, and no additional information is provided in the Materials & Methods (despite the callout).

- p16 and Figure 5 - Reduction of the Cell Painting profiles to a single number (indicating the lowest concentration at which a “strong biological fingerprint was observed”) masks what one presumes to be a tremendous amount of information on the bioactivity of these compounds. It is important that they are not only active, but specific, and more analysis of and information on the profiles should be used to assess this question. Further, no primary data are presented in the Supporting Information. What constitutes a “strong biological fingerprint”? This represents a major gap in the presentation and data.

- p17 - The authors should be very careful not to overstate their results on antibacterial activity. The MIC values are modest at best (32, 64, or even 128 uM). Indeed, >32 is often used as a cutoff for ‘inactivity’ in studies of antibacterial compounds. Even in the pore-overexpression E. coli strain, a high concentration of 50 uM is used to achieve the %inhibition reported. This part of the presentation should be revised to be more circumspect.

- p17 - “Similarity between antibacterial profiles and selectivity between pore overexpression strains suggests that these molecules share a common mode of action, rather than merely sharing similar physicochemical properties (and therefore similar uptake profiles).” The logic here is confusing. Why would a similar profile not simply imply similar uptake? What specific compound profiles are being referred to?

- Rigor (p19) - There is extensive discussion of the hits from the library screen, toward the overarching claim that the library is rich in biological activity. Were the hits were QCed to confirm identity and retested to confirm activity? Were the diastereomers separated to assess individual activities? Were enantiomers analyzed to assess specificity? The latter is critical to demonstrating that the compounds are not only bioactive but have specific mechanisms of action.

- p19 (and Introduction p7) - The claim of a 12% hit rate in comparison to conventional HTS hit rates of 0.1-0.5% is misleading. Most HTS assays are conducted against a single target or phenotype. In calculating their 12% hit rate, the authors have aggregated data from multiple different assays (>17!). To enable an apples-to-apples comparison, a negative control screen must be done using a conventional HTS library of comparable size. This would be the preferred approach as it would be more informative and interesting scientifically; alternatively, however, the authors may also revise their presentation to remove the 12% claim, which cannot be compared to other results. (see also p21: "hit rates that far exceed those typically encountered")

Other issues:

- Abstract - the authors claim that the library is of "unprecedented complexity and size", but later cite Schreiber's pioneering work as, in effect, precedent (p5).
- Figure 1C - this panel is confusing, too dark, and the structures are too small to read. The x-axis labels suggests that the distinguishing feature of natural products is solely their complexity, but this does not seem to be a key concept demonstrated in the paper, and oversimplifies the distinguishing characteristics of natural products identified in the field over the last 2 decades.
- Figure 2A - the fragments shown do not follow from the exemplar natural product shown; either more examples of natural products should be shown, or the fragments limited to those derived from the specific exemplar.
- Figure 2B - it would be helpful to color code the three fragments in the exemplar library member.
- Figure 2F - a maximum Cs of 0.70 is indicated (" $0.70 > Cs > 0.55$ "), but the rationale for or relevance of this cutoff is not discussed in the main text.
- p9 - Use of the term "training set" to describe the 18 natural products implies that a machine learning approach was used, which is not the case. In other places, the term "reference set" is used, which is more appropriate here, as the structural deconstruction was manual.

- p13 - Figure 4 should be called out here when the complete library is described (7-18a-c). (Otherwise, it is confusing because these compound numbers do not appear in Figure 3.)
- Figure 4 - Compound number 10d is used to describe two different compounds. The rearranged structures currently annotated with † should be given a separate compound number, or at least designated 10d' to differentiate them.
- Figure 5A - The use of different units (lower is better for MIC, but higher is better for % inhibition) makes the table confusing. The presentation should be revised for clarity.
- Supporting Information - Despite the impressive 442 pages of Supporting Information, information noted above on protocols for pore-overexpression strains and primary data from Cell Painting assays is missing. In addition, NMR data is missing for the acetate derivatives (50 compounds).

Reviewer #3 (Remarks to the Author):

The paper is an excellent addition to the literature and is a strong mix between synthesis, modelling, and informatics. I only have a few comments.

“Polyketide macrolides (pMLs) have long been a source of inspiration for drug discovery” can you provide specific examples where non-natural product drugs were, in fact, designed this way?

“Family of THF-containing macrolides were deconstructed?” I understand what this means, but still I feel that the term needs to be defined better.

The notion of conformational overlap against natural products is also not described in detail. I feel that this is ill-defined.

“each conformer within 15 kcal/mol of the global minimum for that compound was retained”: where is this 15 kcal/mol cut-off from? It is not clear.

The molecules contain lactone linkages and I do not see these to be particularly stable. The authors need to comment on that.

We thank the three reviewers for providing insightful comments on our manuscript entitled “Targeted Sampling of Natural Product Space to Identify Bioactive Natural Product-Like Polyketide Macrolides”. We have addressed all comments from the reviewers in the attached revised manuscript (all changes highlighted in yellow) and have provided responses to each comment below.

Reviewer #1 (Remarks to the Author):

Britton et al. describe targeted sampling of natural product space as a new molecular design concept to explore focused yet prevalidated areas of biologically relevant chemical space. Highly complex, yet highly biologically relevant tetrahydrofuran-containing polyketide macrolides were chosen as starting points and were deconstructed into fragments and reassembled with all possible combinations in silico to afford >3000 virtual molecules. Conformational and functional group overlap was scored and virtual compounds that had the lowest similarity relative to the guiding NPs were prioritized for synthesis. A convergent synthetic approach utilizing the groups expertise in macrolide synthesis was used to access 170 compounds (including acetylated derivatives) from a small set of unique building blocks. Biological performance of the collection was evaluated by BioMAP and morphological profiling, revealing substantially higher hit rates than typical screening collections and adding support for the design principle as a means to efficiently access novel biologically active chemical matter.

Overall, this is a nice piece of work that has rationally and efficiently given access to novel structures that lie in an area of biologically-rich chemical space. The conclusions are consistent with the data presented, the manuscript is well written, and the methods and compounds are well characterized in the supporting information. Of note, it is particularly impressive that the designed and synthesized compounds retain the complexity of the parent natural products, which is not usually the case for collections inspired by polyketide macrolides. This is an important contribution to the field, and I would recommend this work for publication in Nature Communications after addressing these minor points.

Response: We thank the Reviewer for their kind and supportive comments and, particularly, for noting the unique complexity of the targeted polyketide macrolides.

Comment 1. The introduction mentions that macrocycles typically show poor adherence to conventional rules for drug-likeness but yet are able to enter cells and be successful drugs. Some explanations are given how they can overcome these obstacles, but it would be beneficial elaborate on the possibility of solute carrier proteins as an alternative to passive diffusion (Nat. Rev. Drug Discov. 2008, 7, 205-220). This is especially relevant for this study as the compounds tested are highly natural product-like and may therefore have recognition elements for transport proteins.

Response: We appreciate the Reviewer pointing out this manuscript. We have added the following sentence along with a reference (reference 8) to this manuscript: “Macrocycles along with many other natural products may also benefit from active transport mechanisms.⁸”

Comment 2. The proposed TSNaP approach has several similar features to the pseudo-natural product design concept (J. Am. Chem. Soc. 2022, 144, 3314). Both concepts deconstruct

natural products into natural product fragments and recombine them in different arrangements. However, the pseudo-natural product concept provides cyclic fragments whereas the TSNaP approach can provide both cyclic and acyclic fragments. TSNaP may be complimentary to pseudo-natural product design and, as demonstrated, may be much more useful when employing macrocyclic natural products as starting points. A brief comparison between the two methods should be included in the introduction.

Response: We appreciate this comment as well as a related comment by Reviewer 2. We have added the following statements to the introduction as well as references to the manuscripts noted below to provide a comparison between the two methods.

“The TSNaP approach builds on other strategies used to target natural product-relevant chemical space^{30,31} and has similarities to pseudo-natural product design pioneered by Waldmann and co-workers.^{32–34} In both TSNaP and pseudo-natural product design, compounds are synthesized using NP inspired synthetic fragments. However, unlike pseudo-natural product design, TSNaP prioritizes whole molecules rather than fragments on the basis of their calculated structural similarity to a reference set of bioactive natural products. In principle, TSNaP can be applied to any sufficiently related family of known bioactives comprised of cyclic and acyclic fragments.”

30. Koch, M. A. *et al.* Charting biologically relevant chemical space: a structural classification of natural products (SCONP). *Proc. Natl. Acad. Sci.* **102**, 17272–17277 (2005).
31. van Hattum, H. & Waldmann, H. Biology-oriented synthesis: harnessing the power of evolution. *J. Am. Chem. Soc.* **136**, 11853–11859 (2014).
32. Karageorgis, G., Foley, D. J., Laraia, L. & Waldmann, H. Principle and design of pseudo-natural products. *Nat. Chem.* **12**, 227–235 (2020).
33. Grigalunas, M., Brakmann, S. & Waldmann, H. Chemical evolution of natural product structure. *J. Am. Chem. Soc.* **144**, 3314–3329 (2022).
34. Niggemeyer, G. *et al.* Synthesis of 20-membered macrocyclic pseudo-natural products yields inducers of I κ B lipidation. *Angew. Chem.* **134**, e202114328 (2022).

Comment 3. Please provide a general range of diastereoselectivities for the Nozaki-Hiyama-Kishi reactions in the main text.

Response: We have edited the manuscript to include the following statement: " In all cases a mixture of epimeric alcohols was produced with diastereomeric ratios ranging from 1:1 to 10:1 at the newly formed hydroxymethine stereocenter. Where possible, the diastereomeric alcohols were separated, however, in many cases these compounds proved to be inseparable by flash column chromatography (as indicated in Figure 3)."

Comment 4. Figure 4 - I would suggest using R groups for the side chains to make the structures more representative.

Response: We have updated Figure 4 to include colored side chains and we have added a legend for clarity. We feel this change makes it clear that each indicated compound is a representative example of each macrocycle chemotype.

Comment 5. Figure 4 – The dots in the outer region (0.65-0.7) are colored orange but should be colored cyan to match the legend.

Response: We apologize for this error and have now corrected this.

Comment 6. Figure 4 – The collection was prepared as a mixture of diastereomers from the NHK reaction. Were both diastereomers considered for the C_s scores or just one diastereomer? Please clarify this in the text and/or in the caption of Figure 4.

Response: In all cases, both diastereomers were considered for the C_s scores. To better communicate this point we have edited the caption of Figure 2F to read: “In the indicated example, three out of the six candidate pMLs had a C_s score between 0.55 and 0.70 and therefore all six compounds are considered targets for synthesis. Finally, the subset comprised of candidate structures was manually refined for structural diversity and minimal synthesis effort resulting in 78 THF pMLs prioritized for synthesis.”

Additionally, we have made the following changes to the text: “Critically, this process enabled prioritization of macrocycles that most closely resembled those found in the natural product reference set (maximum observed $C_s \sim 0.70$), which we hypothesized would help retain biological relevance. In most cases, only a single epimer of a single macrocycle-side chain combination (*e.g.* amide) mimics the natural product reference set ($C_s > 0.55$), while the other two (*i.e.* thiazole, dihydropyran) serve to diversify the backbone (and consequently tend to have lower C_s scores), enabling a broader sampling of THF pML chemical space (**Figure 2F**, supplementary **Figures S4-S6**).”

Comment 7. The caption in Figure 5 states ‘indicate the lowest concentration (μM) at which a strong biological fingerprint was observed’. In the context of the cell painting assay, what does this mean? Is it purely qualitative or is there a quantitative threshold? Please explain this in the caption.

Response: We use the ‘Activity Score’ as a measure of minimum activity. This is defined as the square root of the sum of the squares for the values in the fingerprint. Concentrations are defined as active if the activity score is 5-times greater than the mean activity score for the negative control wells from each plate. We have included this definition in the figure caption, and have generated new plots of activity score vs. concentration for all compounds with activity in the CP assay (Figures S12B – S19B).

Comment 8. Figure 5b – Could you provide comparative CPA fingerprints to show that 7bAc and BML-210 are similar (or to other apoptosis-inducing compounds)?

Response: we have added a new panel to Figure 5 that includes the fingerprints for 7bAc, BML-210, EPZ005687 and other compounds with related fingerprints. We have also expanded the discussion of this cluster by the addition of the following paragraph in the main text:

“The final activity class (red group, **Figure 5A**) contained five compounds (**11bAc, 7e, 7bAc, 12bAc, 18bAc**) which were active in the Cell Painting assay, yielding morphological profiles that were closely related and which grouped with reference compounds (**Figure 5B**) resulting in apoptotic cell death (**Figure 5C**). The reference compounds in this cluster are all involved in the disruption of histone function, including inhibitors of histone acetylation (PFI-4, GSK5959), deacetylation (BML-210, pracinostat, PAOA, UF010, mocetinostat), methylation (EPZ005687) and demethylation (JIB-04). Biselide A and fijianolides A and B have been shown to influence tubulin dynamics, which can be affected by histone function suggesting a possible overlapping role for these four macrolide scaffolds.”

Comment 9. Page 14 – ‘(low overlap, moderate NP-likeness)’, ‘(high overlap, closer similarity to NP scaffolds)’, and page 20 ‘high natural product-likeness)’. I believe the intentions are to compare to reference NPs not NPs in general. Additionally, Cs is not comparing scaffolds but 3D-structural overlap. These should be rephrased to resemble more closely what the comparisons are.

Response: We thank the Reviewer for pointing this out. We have now replaced all instances of “NP-likeness” with “NP reference set-likeness”.

Reviewer #2 (Remarks to the Author):

Linnington, Britton and coworkers report a fragment-based approach to the design of natural product-like libraries. The approach is guided by insights into key structural features of polyketide macrolide natural products and deconstruction into synthetically-accessible fragments, which then can be recombined into natural product-like macrocycles. There are several attractive features of this work, including the macrolide target class, computational approach to library design, descriptions of reaction troubleshooting, size of the library, and demonstration of biological activity. However, the manuscript is also unclear in some places and further makes several expansive claims that are not sufficiently supported by the data. As such, while the work is, in principle, appropriate for publication in Nature Communications, numerous significant revisions are required prior to publication.

Response: We appreciate the comments of the Reviewer as well as their concerns regarding clarity. As detailed below, we have revised the manuscript extensively to address the Reviewers comments.

Comment 1. Scholarship (p6) - Extensive previous work by other groups, particularly Waldmann, on natural product fragment-based libraries is not cited. The senior authors should do a comprehensive literature search, cite the most relevant prior art in the field, and revise the introduction to place the current work in proper context.

Response: A similar comment was made by Reviewer 1 (Comment 2) and we have modified the introductory section of the manuscript to include additional discussion and several new references, including three to Waldmann's excellent studies on natural product fragment-based libraries.

Comment 2. p9 - The approach considers all conformers within 15 kcal/mol of the global minimum for each compound. This is a massive range, the rationale for which is not explained sufficiently in the text. Are such high-energy conformations even accessible in the context of the binding pose with a macromolecular target?

Response: The 15 kcal/mol cut-off is based on comparison of computed lowest energy conformations to X-ray structures of macrocycles bound to proteins. Here, the best fitting geometry can be found at up to 15 kcal/mol relative to its global minimum (see reference 36). Additionally, classical force field methods struggle to accurately order conformational energies. It is thus not uncommon to find an energy minimized conformation that is 10 kcal/mol above the putative minimum via molecular mechanics, which when re-evaluated using DFT methods, turns out to be a low energy conformation (or the global minimum). In our case, DFT calculations on this many compounds/conformations was not possible and therefore, we opted for a conservative energy window that would capture both low/and high energy conformations that may be responsible for binding with proteins.

For clarity, we have edited the relevant section of the manuscript to read: "To ensure that all biologically relevant conformations of a given compound were considered, each conformer within 15 kcal/mol of the global minimum for that compound was retained based on studies by Chen and Foloppe who found this to be a relevant energy window for flexible macrocycles.³⁶"

Comment 3. p9 - "This analysis allowed us to prioritize compounds for synthesis that were sufficiently dissimilar to the natural products training set ($C_s < 0.55$), but that contained at least one macrocycle side-chain analogue with relatively greater similarity ($C_s > 0.55$)." This statement is confusing as it implies that the side chains were analyzed separately from the complete structures. Only after reading this and Figure 1F several times did it become clear that the authors are referring to sets comprised of a single scaffold with the 3 different side chains. The $C_s < 0.55$ criteria is also not shown in Figure 1F nor stated in the caption. This presentation needs to be revised.

Response: We appreciate this comment, which was also raised by Reviewer 1. We have addressed this concern in the response to Comment 6, above.

Comment 4. Figure 3 - From the total library of 3456 possible structures, the authors selected a subset of building blocks for synthetic feasibility to target 78 library members. However, the 5 x 8 x 3 building blocks selected provides a maximum of 120 combinatorial structures. The authors should comment on why the other 42 structures were not selected for synthesis.

Response: As noted above (Reviewer 1, Comment 6), in the final stage, we prioritized based on synthetic feasibility. Ultimately, this final prioritization resulted in our targeting of 12 distinct total syntheses of pML chemotypes.

Comment 5. p13 - The production of diastereomeric mixtures detracts from the elegance of the library design and synthesis, especially as the compounds are screened as mixtures that may have independent biological activities. Did the authors investigate any asymmetric or diastereoselective alternatives to the NHK coupling? What is the dr? Are the diastereomers separable?

Response: We appreciate this comment from the Reviewer and have addressed this in part in our response to Reviewer No. 1 (Comment 3). We did not undertake the development of separate methodologies for asymmetric NHK reactions. While we also have experience with these processes, we have found them to be highly substrate dependent and would have been difficult to implement with 12 distinct macrolide cores and 3 distinct sidechains.

Comment 6. p13 - Acetylation of the library increases the number of compounds produced, but these compounds will certainly be metabolically unstable in vivo (in animals). The authors cite literature on potential utility in vitro (cell-based assays), but do not note to this major downstream liability.

Response: We thank the Reviewer for this comment and note that our goal for this proof-of-concept study was to identify compounds that had potentially useful biological activity in cellular assays that could serve as leads for drug discovery. Thus, we aimed to improve cellular permeability to support this goal and were not focused on in vivo activity. We have added an additional and, we feel, valuable reference relating to active transport of natural products and natural product-like compounds that was suggested by Reviewer No. 1.

Comment 7. Figure 4 - There appear to be 48 non-acetylated compounds shown, with 47 acetylated compounds, for a total of 95 structures. If diastereomers are counted separately, there would be $96 + 94 = 190$ structures. Yet, the library size is stated as 170 compounds; this needs to be clarified.

Response: For compounds containing a side chain there are two diastereomers. Macrocyclic core compounds are a single compound, and the thiazole ketone is a single compound.

Additionally, as depicted in Figure 4, compound 18 is a mixture of 4 diastereomers (E/Z isomers as well as R/S isomers). We also note that the thiazole side chain on compound 9 (i.e., 9c) proved to be unstable as was the macrocyclic core for compound 11 (i.e., 11d) and are not included in Figure 4. Finally compounds indicated with a dagger in Figure 4 underwent rearrangement as indicated and are not included in the count of pMLs. The following caption now accompanies Figure 4 for clarity. "... † isolated as the translactonization (5 or 6 membered lactone) products, which were not included in the total pML library count."

To be clear regarding the compound count, we have tabulated them below:

7a – 2 compounds

7b – 2 compounds

7c – 2 compounds

7d – 1 compound

7e – 1 compound

7 (total compounds) = 2+2+2+1+1 = 8 compounds

8a – 2 compounds

8b – 2 compounds

8c – 2 compounds

8d – 1 compound

8 (total pMLs) = 2+2+2+2+1 = 7

9 (total compounds) = 5

10 (total compounds) = 5

11 (total compounds) = 4

12 (total compounds) = 7

13 (total compounds) = 7

14 (total compounds) = 7

15 (total compounds) = 7

16 (total compounds) = 7

17 (total compounds) = 7

18 (total compounds) = 14

Total number of non-acylated compounds: 8+7+5+5+4+7+7+7+7+7+7+14 = 85

All compounds were acylated to provide 85 additional acylated pMLs for a total of 85+85 = 170 THF pMLs.

Comment 8. p16 - The use of Gram-negative pore-overexpression strains is mentioned, but the rationale for this is not described in sufficient detail, there are no references cited, and no additional information is provided in the Materials & Methods (despite the callout).

Response: We have edited the manuscript to include the following rationale: “both wild-type and pore-overexpression/efflux-deficient strains^{56,57} engineered to allow improved compound uptake and monitoring of target engagement within bacterial cells (see supplementary information). The direct comparison of the wild-type and pore-overexpression/efflux deficient strains reflects the potential of individual molecules to penetrate through the bacterial outer-membrane and to assess properties to be modulated in order to improve the overall antibacterial activity.”

We have added the following additional references to accompany these statements:

56. Krishnamoorthy, G. *et al.* Breaking the permeability barrier of *Escherichia coli* by controlled hyperporination of the outer membrane. *Antimicrob. Agents Chemother.* **60**, 7372–7381 (2016).

57. Hu, Z. *et al.* Structure–uptake relationship studies of oxazolidinones in gram-negative ESKAPE pathogens. *J. Med. Chem.* **65**, 14144–14179 (2022).

Comment 9. p16 and Figure 5 - Reduction of the Cell Painting profiles to a single number (indicating the lowest concentration at which a “strong biological fingerprint was observed”) masks what one presumes to be a tremendous amount of information on the bioactivity of these compounds. It is important that they are not only active, but specific, and more analysis of and information on the profiles should be used to assess this question. Further, no primary data are presented in the Supporting Information. What constitutes a “strong biological fingerprint”? This represents a major gap in the presentation and data.

Response: The question of how strong biological fingerprints were defined was also asked by reviewer 1 and has been addressed in comment 7 above. We have also added a new set of plots depicting the activity scores as a function of concentration for each active compound in the table.

To address the issue of raw data presentation we have included a set of csv files containing the processed numerical fingerprints (~1,000 metrics per row) for the dilution series for each active compound in table 5A as a zip file in the supporting information. This file also includes the fingerprint data for fijianolides A and B. Each file includes a header row that defines the metrics that were extracted from the raw images.

To address the issue of specificity we re-evaluated the CP activity class discussed in the manuscript (red group, figure 5A – C). We re-examined the published biological mechanisms ascribed to all reference compounds in this cluster and created an expanded paragraph

discussing the relationship between reported mechanisms, and the implications for compound 7bAc. This new text is reproduced in the response to reviewer 1, comment 8. Finally, we have added hierarchical clustering plots for dilution series of relevant compounds in the supporting information (Figures S12A-S19A).

Comment 10. p17 - The authors should be very careful not to overstate their results on antibacterial activity. The MIC values are modest at best (32, 64, or even 128 μM). Indeed, >32 is often used as a cutoff for ‘inactivity’ in studies of antibacterial compounds. Even in the pore-overexpression *E. coli* strain, a high concentration of 50 μM is used to achieve the %inhibition reported. This part of the presentation should be revised to be more circumspect.

Response: We have reviewed this section of the discussion and revised the text to more clearly state the moderate potencies and removed statements about unique spectrum of activity for these classes, given that activities are close to the top tested concentrations in some cases. We have also explicitly stated the concentration used for *E. coli* pore overexpression strain testing in the main text.

Comment 11. p17 - “Similarity between antibacterial profiles and selectivity between pore overexpression strains suggests that these molecules share a common mode of action, rather than merely sharing similar physicochemical properties (and therefore similar uptake profiles).” The logic here is confusing. Why would a similar profile not simply imply similar uptake? What specific compound profiles are being referred to?

Response: It is tempting to propose that biological activity profiles in multi-organism panels are indicative of MOA (in other words that compounds with similar MOAs should afford similar activity profiles). However, this assumption ignores the fact that different scaffolds will have different uptake/efflux properties in different organisms in the panel. Therefore, compounds with different structures but similar molecule targets (e.g. erythromycin and chloramphenicol, both of which disrupt the entry or exit of transfer RNA (tRNA) in the peptidyl transferase center (PTC) of the ribosomal 50S subunit) will have very different profiles in the BioMAP panel. This distinction is reduced in pore overexpression systems because uptake/efflux does not influence activity to the same extent that it does in WT strains. Therefore it is reasonable to suggest that similar activity profiles in both WT and overexpression panels is indicative of shared MOA rather than similar uptake/efflux profiles for compounds with different mechanisms of action.

We have modified the discussion in this paragraph as follows:

“Interestingly, many of the compounds in this group were also active against the pore overexpression strain of *E. coli* at 50 μM but were not active against the other pore overexpression strains or the corresponding WT *E. coli* strain. Similarity between antibacterial profiles in the WT BioMAP panel and selectivity between organisms in the pore overexpression panel suggests that these molecules may share a common mode of action,

rather than merely possessing similar physicochemical properties (and therefore similar uptake profiles) for compounds with differing MOAs.”

Comment 12. Rigor (p19) - There is extensive discussion of the hits from the library screen, toward the overarching claim that the library is rich in biological activity. Were the hits were QCed to confirm identity and retested to confirm activity? Were the diastereomers separated to assess individual activities? Were enantiomers analyzed to assess specificity? The latter is critical to demonstrating that the compounds are not only bioactive but have specific mechanisms of action.

Response: All library members were screened as dilution series in all assays. This approach affords dose-response data for each hit in the assay, which acts as an internal validation for primary screening data. All compounds were produced as single enantiomers as indicated in the synthesis section. We did not carry out synthesis of the enantiomeric library of compounds. All compounds discussed in the manuscript possess dose-dependent activities, with the exception of a few compounds in the brown group in Figure 5 that were only active at the top tested concentration. All compounds in the original library were identified using extensive 1D and 2D NMR analyses, coupled with HRMS analyses. All acetylated products were ‘spot-to-spot’ conversions that were performed on 1 mg scale and confirmed by HRMS analyses. Where possible, diastereomers were separated and screened as individual compounds, as discussed in the response to Reviewer 1 comment 3 and reviewer 3 comment 23.

Comment 13. p19 (and Introduction p7) - The claim of a 12% hit rate in comparison to conventional HTS hit rates of 0.1-0.5% is misleading. Most HTS assays are conducted against a single target or phenotype. In calculating their 12% hit rate, the authors have aggregated data from multiple different assays (>17!). To enable an apples-to-apples comparison, a negative control screen must be done using a conventional HTS library of comparable size. This would be the preferred approach as it would be more informative and interesting scientifically; alternatively, however, the authors may also revise their presentation to remove the 12% claim, which cannot be compared to other results. (see also p21: “hit rates that far exceed those typically encountered”)

Response: We have removed the statements regarding hit rate and modified the discussion to highlight the proportion of library pMLs that were active in one or more of the panel of assays we screened against. The manuscript now states: “As highlighted here, this approach resulted in approximately 12% of the targeted library pMLs displaying activity in one or more of the assays evaluated, and produced compounds that closely resemble naturally occurring pMLs but exhibit different activities from the natural products that inspired them.”

And: Overall, approximately 12% (21 out of the 170 synthesized THF pMLs) showed strong biological fingerprints in one or more of our panel of biological assays.”

Comment 14. Abstract - the authors claim that the library is of “unprecedented complexity and size”, but later cite Schreiber’s pioneering work as, in effect, precedent (p5).

Response: We have removed “unprecedented complexity and size”. The text now reads: “Using a modular and stereoselective synthetic approach, a library of polyketide-like macrolides was then prepared to sample these unpopulated regions of pML chemical space.”

Comment 15. Figure 1C - this panel is confusing, too dark, and the structures are too small to read. The x-axis labels suggests that the distinguishing feature of natural products is solely their complexity, but this does not seem to be a key concept demonstrated in the paper, and oversimplifies the distinguishing characteristics of natural products identified in the field over the last 2 decades.

Response: Figure 1C has now been revised significantly. It has been lightened, the structure size increase, and we have emphasized similarity to the NP reference set instead of ‘complexity’ to increase both clarity and scientific accuracy.

Comment 16. Figure 2A - the fragments shown do not follow from the exemplar natural product shown; either more examples of natural products should be shown, or the fragments limited to those derived from the specific exemplar.

Response: Our intention was to show the NP inspired building blocks rather than direct disconnections. For clarity, these building blocks are now labelled as “NP inspired building blocks”.

Comment 17. Figure 2B - it would be helpful to color code the three fragments in the exemplar library member.

Response: The three fragments are now appropriately color coded throughout Figure 2.

Comment 18. Figure 2F - a maximum C_s of 0.70 is indicated (“ $0.70 > C_s > 0.55$ ”), but the rationale for or relevance of this cutoff is not discussed in the main text.

Response: The maximum C_s of 0.7 was the maximum C_s observed and as a result the upper threshold. To be clear that this was not a cut off, the text has now been edited as follows: “Critically, this process enabled prioritization of macrocycles that most closely resembled those found in the natural product reference set (maximum observed $C_s \sim 0.70$), which we hypothesized would help retain biological relevance. In most cases, only a single epimer of a single macrocycle-side chain combination (*e.g.* amide) mimics the natural product reference set ($C_s > 0.55$), while the other two (*i.e.* thiazole, dihydropyran) serve to diversify the backbone (and consequently tend to have lower C_s scores), enabling a broader sampling of THF pML chemical space (**Figure 2F**, supplementary **Figures S4-S6**).”

Comment 19. p9 - Use of the term “training set” to describe the 18 natural products implies that a machine learning approach was used, which is not the case. In other places, the term “reference set” is used, which is more appropriate here, as the structural deconstruction was manual.

Response: We apologize for this misuse of “training set”. Training set has now been replaced with “reference set” throughout.

Comment 20. p13 - Figure 4 should be called out here when the complete library is described (7-18a-c). (Otherwise, it is confusing because these compound numbers do not appear in Figure 3.)

Response: The text has been revised as follows: “Despite this additional complexity, the majority (72) of the 78 targeted compounds **7-18a-c** (each a mixture of 2 diastereomers at point of side chain attachment) were successfully prepared following this strategy along with 12 macrocycles lacking a side chain (**Figure 4**).”

Comment 21. Figure 4 - Compound number 10d is used to describe two different compounds. The rearranged structures currently annotated with † should be given a separate compound number, or at least designated 10d' to differentiate them.

Response: We apologize for this oversight and Figure 4 has now been corrected.

Comment 22. Figure 5A - The use of different units (lower is better for MIC, but higher is better for % inhibition) makes the table confusing. The presentation should be revised for clarity.

Response: We agree that this presentation is somewhat confusing. Unfortunately, this is a consequence of the designs of the different assays. BioMAP assays were configured to measure growth across a dilution series for each compound, permitting the determination of MIC values in all cases. By contrast, the overexpression pore screens were performed at a single fixed dose (50 μ M) for each compound. Therefore, the raw data does not contain sufficient information to determine MIC values for the overexpression screens. The figure caption explicitly states the units for each screen, which we hope will aid readers in interpreting these data.

Comment 23. Supporting Information - Despite the impressive 442 pages of Supporting Information, information noted above on protocols for pore-overexpression strains and primary data from Cell Painting assays is missing. In addition, NMR data is missing for the acetate derivatives (50 compounds).

Response: The primary data from the Cell Painting Assays have been added to SI and (in part) to Figure 5. Protocols for the antimicrobial susceptibility testing of the wild type and pore-overexpressing/efflux-deficient strains have been added to the SI.

With respect to the acetate derivatives the *Nature Communications* author guidelines state that: “Authors describing the preparation of combinatorial libraries should include standard characterisation data for a diverse panel of library components.” Here, we rigorously characterized all of the individual building blocks (and key intermediates in their synthesis), and fully characterized all final macrocyclic compounds prior to acetylation. Additionally, we characterized key intermediates en-route to final compounds. All acetylation reactions were quantitative and “spot-to-spot” transformations as noted by TLC analysis. As we only prepared 1 mg of acetylated macrocycles we relied on HRMS of the purified compounds. As noted by the Reviewer, the characterization data is impressive and we feel sufficient.

Reviewer #3 (Remarks to the Author):

The paper is an excellent addition to the literature and is a strong mix between synthesis, modelling, and informatics. I only have a few comments.

Response: We thank Reviewer 3 for their strong and supportive comments.

Comment 1. “Polyketide macrolides (pMLs) have long been a source of inspiration for drug discovery” can you provide specific examples where non-natural product drugs were, in fact, designed this way?

Response: We have added references to representative examples of drugs in this category, including eribulin and moxidectin.

14. Yu, M. J., Zheng, W. & Seletsky, B. M. From micrograms to grams: scale-up synthesis of eribulin mesylate. *Nat. Prod. Rep.* **30**, 1158–1164 (2013).

15. Milton, P., Hamley, J. I. D., Walker, M. & Basáñez, M.-G. Moxidectin: an oral treatment for human onchocerciasis. *Expert Rev. Anti Infect. Ther.* **18**, 1067–1081 (2020).

Comment 2. “Family of THF-containing macrolides were deconstructed?” I understand what this means, but still I feel that the term needs to be defined better.

Response: The text has been revised to address this comment as follows: “Using TSNaP, a family of tetrahydrofuran-containing pMLs were computationally assembled from pML inspired building blocks to provide a large collection of natural product-like virtual pMLs.”

Comment 3. The notion of conformational overlap against natural products is also not described in detail. I feel that this is ill-defined.

Response: We apologize for this lack of clarity. Conformational overlap was meant to indicate the volumetric overlap of two individual conformations (the formal terminology for the term

calculated). We have replaced the vague term “conformational overlap” with the more robust (and accurate) “volumetric overlap”.

Comment 4. “each conformer within 15 kcal/mol of the global minimum for that compound was retained”: where is this 15 kcal/mol cut-off from? It is not clear.

Response: A similar comment was made by Reviewer 2 (Comment 2) and is addressed above.

Comment 5. The molecules contain lactone linkages and I do not see these to be particularly stable. The authors need to comment on that.

Response: Macrolide ester linkage stability is highly dependent on ring size and molecular conformation. For example, almost all of the macrolide class antibiotics (erythromycin etc.) contain lactone ring closures yet are in regular use in the clinic. While we did not assess liver microsomal stabilities for library members, the list of FDA-approved drugs includes many examples of lactone-bearing macrolides (e.g. ivermectin, clarithromycin etc.). We have included an additional sentence in the discussion of the library design discussing the inclusion of the lactone moiety.

REVIEWER COMMENTS

Reviewer #1 (Remarks to the Author):

The authors have fully addressed my comments, and I would recommend the revised version of this manuscript for publication in Nature Communications.

Reviewer #2 (Remarks to the Author):

In this revised manuscript, Linnington, Britton, and coworkers have ably addressed most of concerns raised by myself and the other two reviewers. Two responses in the rebuttal bear further discussion:

1) On the question of biological specificity (labeled Comment 12 in the rebuttal), analysis of an enantiomeric compound is a standard approach to addressing this issue, as enantiomers have identical physicochemical properties but often interact very differently with biological targets, which present chiral binding sites. This analysis does NOT require “synthesis of the enantiomeric library” as the authors state in their rebuttal. It only requires synthesis of enantiomers of the hits. As the goal of the work presented is to assess the overall biological activity profile of the library, I am inclined to set this point aside, but the authors would be expected to conduct this analysis in any future studies focusing on individual bioactives.

2) Regarding characterization of the acetates (labeled Comment 23 in the rebuttal), the Nature Communications guideline clearly requires characterization of a representative subset of library members. The alcohols certainly can not be considered representative of the acetates. The “spot-to-spot” TLC analysis cited in the rebuttal is not rigorous or quantitative and would not be accepted as definitive in any chemistry journal. Likewise, HRMS does not define connectivity and would not, for example, be able to distinguish between rearrangement products such as allylic acetates. One mg should be sufficient for high-field NMR analysis (especially for a “spot-to-spot” transformation), or the authors can scale up a representative subset of acetates to facilitate characterization. The structures claimed in this paper will be indexed in CAS, and thus, their structures must be confirmed, at least for a representative subset of acetates. Failure to sufficiently characterize half of the compounds claimed in the manuscript, which represent 2/3rds of the hits in Figure 5A, threatens to undermine what is otherwise an excellent contribution to the field.

REVIEWER COMMENTS

Reviewer #1 (Remarks to the Author):

Comment: The authors have fully addressed my comments, and I would recommend the revised version of this manuscript for publication in Nature Communications.

Response: We thank Reviewer #1 for their time spent reading this manuscript and for providing constructive comments.

Reviewer #2 (Remarks to the Author):

In this revised manuscript, Linnington, Britton, and coworkers have ably addressed most of concerns raised by myself and the other two reviewers. Two responses in the rebuttal bear further discussion:

Comment 1: On the question of biological specificity (labeled Comment 12 in the rebuttal), analysis of an enantiomeric compound is a standard approach to addressing this issue, as enantiomers have identical physicochemical properties but often interact very differently with biological targets, which present chiral binding sites. This analysis does NOT require “synthesis of the enantiomeric library” as the authors state in their rebuttal. It only requires synthesis of enantiomers of the hits. As the goal of the work presented is to assess the overall biological activity profile of the library, I am inclined to set this point aside, but the authors would be expected to conduct this analysis in any future studies focusing on individual bioactives.

Response: We thank Reviewer #2 for the suggestion for future studies.

Comment 2: Regarding characterization of the acetates (labeled Comment 23 in the rebuttal), the Nature Communications guideline clearly requires characterization of a representative subset of library members. The alcohols certainly can not be considered representative of the acetates. The “spot-to-spot” TLC analysis cited in the rebuttal is not rigorous or quantitative and would not be accepted as definitive in any chemistry journal. Likewise, HRMS does not define connectivity and would not, for example, be able to distinguish between rearrangement products such as allylic acetates. One mg should be sufficient for high-field NMR analysis (especially for a “spot-to-spot” transformation), or the authors can scale up a representative subset of acetates to facilitate characterization. The structures claimed in this paper will be indexed in CAS, and thus, their structures must be confirmed, at least for a representative subset of acetates. Failure to sufficiently characterize half of the compounds claimed in the manuscript, which represent 2/3rds of the hits in Figure 5A, threatens to undermine what is otherwise an excellent contribution to the field.

Response: separate correspondence with Dr. Majda Bratovic (Senior Editor) on this comment resulted in the following agreed upon proposal. Additionally, Dr. Bratovic stated that “Aside from these 10 compounds, I would kindly ask you to characterize seven additional compounds to have covered about 10% of your library.”

From email correspondence on Nov 28: “In our view, the request from Reviewer #2 and your team to characterize a 'subset' of the acetates is reasonable but fundamentally different from a request to characterize specific 'hit' compounds. Ultimately, this approach will not result in the characterization of an unbiased subset of the library. As a further concern, the majority of 'hit' acetates belong to the compound class 18a-dAc, which is the most structurally simple family of macrolides and does not adequately reflect the diversity in macrolide chemotypes. Thus, we would appreciate it if you and your team would consider a compromise where we characterize 25% of the acetates (see list below), which represents 75% of the macrolide chemotypes and 30% of all 'hit' acetates. We feel this is a more appropriate 'subset' that better reflects the structural diversity found in the library and better satisfies library characterization requirements.

'Hit' acetates: 7eAc, 12bAc, 14bAc, 18cAc

non 'Hit' acetates: 8aAc, 12cAc, 13aAc, 15cAc, 16bAc, 17cAc

We have now completed the characterization of a subset of the library that includes ~25% of the acetylated macrolides (see list below), which represent 75% of the macrolide chemotypes and 30% of all 'hit' acetates. As proposed in our email correspondence, we have fully characterized the following:

'Hit' acetates: 7eAc, 12bAc, 14bAc, 18cAc

non 'Hit' acetates: 8aAc, 12cAc, 13aAc, 15cAc, 16bAc, 17cAc

All characterization information has been added to the revised Supplementary Information file.

In the above referenced email correspondence, Dr. Bratovic also requested that we characterize 7 additional compounds to have then covered about 10% of our library. We note that the compound numbers listed above as 'Hit' or non 'Hit' acetates each represent 1, 2 or 4 diastereomeric macrolide structures. Thus, the 10 compound numbers listed represent a total of 20 macrolide compounds and 20/85 acetylated macrolides (~24%). We also note that in our original submission we included the full characterization data for all 85/85 (100%) of non-acylated macrolides in the library. As a result, with the current additional characterization data, we have now fully characterized 105/170 or 62% of the entire library of macrolides and trust that this satisfies the requirements for library characterization.

REVIEWERS' COMMENTS

Reviewer #2 (Remarks to the Author):

In this revised manuscript, Linnington, Britton, and coworkers have addressed my remaining concerns about appropriate characterization of the acetate library members. It is unfortunate that the acetate NMR spectra were run in C₆D₆, as this precludes direct comparison to the precursor alcohols, which were run in CDCl₃ or other solvents. Nonetheless, I do believe that the standard of characterization has been met and the authors are congratulated on this excellent contribution to the field.

Reviewer #2 (Remarks to the Author):

In this revised manuscript, Linnington, Britton, and coworkers have addressed my remaining concerns about appropriate characterization of the acetate library members. It is unfortunate that the acetate NMR spectra were run in C₆D₆, as this precludes direct comparison to the precursor alcohols, which were run in CDCl₃ or other solvents. Nonetheless, I do believe that the standard of characterization has been met and the authors are congratulated on this excellent contribution to the field.

Response: We thank reviewer 2 for their time in reading and commenting our manuscript.